# Marketization of Energy Resources in China: An Environmental CGE Analysis

**Li Yang** [1] **and Ya Gao** [2],*

1   School of Public Finance and Taxation, Central University of Finance and Economics, Beijing 100081, China; 2018110013@email.cufe.edu.cn
2   China Center for Information Industry Development, Beijing 100048, China
*   Correspondence: 2018110006@email.cufe.edu.cn

**Abstract:** This study aims to examine the effects of energy price fluctuations on China's energy-environment-economy system under different scenarios. To achieve this, a computable general equilibrium model is constructed using the 2020 macroeconomic SAM table and microeconomic SAM tables that encompass 8 energy sectors and 13 intermediate sectors. The model is utilized to analyze the impacts of various policies on variables within the energy-environment-economy system. The findings indicate that an increase in energy prices will lead to a contraction effect on multiple industrial sectors and the overall macroeconomy. Higher energy prices result in elevated prices, reduced output, decreased investment, and decreased consumer spending across most industrial sectors, negatively affecting the macroeconomy. However, government regulation of secondary energy prices can mitigate the influence of primary energy prices on the national economy. Such regulation hinders the transmission of primary energy price fluctuations to downstream industrial chains, thereby alleviating its impact on different sectors and the macroeconomy to varying extents. In order to mitigate the adverse effects of energy price fluctuations, it is crucial to reduce energy consumption while promoting economic growth and enhancing resident welfare. This paper presents relevant measures and suggestions to address these challenges.

**Keywords:** energy prices; environment; CGE model; policy simulation

## 1. Introduction

Energy resource markets in China have undergone a process of liberalization, while the secondary energy markets still remain predominantly under government control. This transitional experience provides valuable insights not only for understanding ongoing changes in China but also for other developing nations that are yet to fully embrace market-oriented approaches [1]. Consequently, energy price reform has become a central focus of government policies aimed at addressing economic, energy, and environmental challenges. China's energy structure exhibits some distinctive features, with coal playing a particularly significant role. According to BP's energy statistics in 2022, coal accounts for 58.2% of China's energy consumption, which is considerably higher than the global average of 27.2%. Moreover, clean energy sources only make up 22.2% of China's energy consumption, lower than the global average of 39.2%. The prices of production factors such as capital, labor, and energy in China are distorted to varying degrees due to market segmentation, government regulation, monopolistic forces, and other factors. These distortions severely hinder the efficient functioning of the market in optimizing resource allocation [2]. Primarily driven by the goals of maintaining stable economic growth and ensuring people's well-being, energy prices in China continue to be largely controlled by the government, thus failing to fully reflect the scarcity of energy resources and the laws of market supply and demand [3]. In recent years, scholars have increasingly focused on the topic of energy price distortions and their implications for resources and the environment. However, most

existing studies primarily concentrate on empirically assessing the resource and environmental effects resulting from energy price fluctuations [4–9]. In reality, different types of energy exhibit heterogeneity in the production process and substitution relationships during usage. These factors influence the mechanisms through which energy price fluctuations impact the energy-environment-economy system [10]. Furthermore, existing literature does not adequately differentiate between various energy sectors, making it challenging to simulate the impact of different energy price changes on the national economy, particularly considering China's unique circumstances.

This paper introduces several key innovations. Firstly, it constructs an energy price CGE model that distinguishes between primary and secondary energies. Secondly, it examines the impact of primary energy price fluctuations on the national economy from both market mechanisms and government regulation perspectives. Thirdly, it explores the influence of secondary energy price fluctuations on the economy through market mechanisms. By evaluating the variations among different schemes on the environment and economic development, this study offers a more precise and quantitative analysis of the effects of energy price fluctuations.

The structure of this paper unfolds as follows: Section 2 provides a literature review. Section 3 establishes the CGE model of energy prices and SAM tables. In Section 4, we simulate various energy price policies, and finally, Section 5 concludes with recommendations and insights.

## 2. Literature Review

Energy prices, such as coal and oil, are quantifiable and exhibit time-series traits. This has led to substantial research using econometric models to study the economic impact of energy price fluctuations. These studies offer valuable insights into the short and long-term economic effects of these fluctuations. Various econometric models like VAR, SVAR, ECM, etc., have been utilized in this context. Kim et al. extended the regional econometric input-output model [11], while Kratena et al. simulated European resource usage scenarios [12]. Istemi et al. examined the influence of energy prices on economic growth [13], and Magali explored the correlation between energy prices and the real effective exchange rate of commodity-exporting countries [14]. However, despite their usefulness, econometric models require stable historical data and often lose critical information during data processing. Their empirical results are sensitive to variable selection and subjective model settings. Furthermore, they only offer trend-based analysis of the impact of energy price fluctuations. Hence, CGE models are considered more appropriate for studying the impact of energy prices [15,16].

The categorization of the energy sector is a critical aspect in this type of research. He et al. segmented the energy sector into five divisions: coal mining, natural gas, oil mining and smelting, coke, and electricity and heat production and supply, to analyze the impact of rising coal prices on electricity prices and macroeconomic variables [17]. Li et al. divided the energy sector into six parts: coal mining, oil and natural gas mining, oil refining, coking and nuclear fuel processing, electricity and heat production and supply, and gas production and supply, to discuss the economic impact of imposing a carbon tax of 100 yuan/ton under varying electricity pricing mechanisms [18]. Liu et al. classified the energy sector into four segments: coal, crude oil, oil refining, and electricity, to study the interplay and response between oil price fluctuations and monetary policies and objectives [19]. Dong et al. divided the energy sector into four segments: coal, oil, natural gas, and electricity, to examine the impact of international oil price shocks and changes in the Renminbi exchange rate on China's macroeconomy [20]. Zhang et al. categorized the energy sector into four divisions: coal, oil, natural gas, and electricity, to analyze the impact of natural gas price fluctuations on the economic system [21]. Lastly, He et al. segmented the energy sector into six departments: natural gas, crude oil, coal, oil, gas, and electricity, to discuss the economic impact of natural gas price fluctuations [22].

In summary, it's evident that foreign researchers primarily focus on the impact of oil price fluctuations on the economy, due to their more developed market economies and different energy consumption structures compared to China. Most domestic scholars still predominantly employ traditional econometric models and input-output methods in this field, with only a few utilizing CGE models. Given China's relatively recent transition to a market economy and its unique national conditions, energy prices have not been fully liberalized, particularly secondary energy prices which remain government-controlled. Therefore, it's crucial to tailor research to China's specific energy market conditions. China's market economy, being only about 30 years old, significantly differs from those abroad. A key characteristic is the government control over China's energy market, especially the secondary one. In the long term, China's energy market will undoubtedly become market-oriented. Hence, it is necessary to study primary and secondary energy separately. Understanding the varying impacts of energy price fluctuations under government regulation and market mechanisms on the economy holds significant relevance for policy formulation.

### 3. CGE Modeling and SAM Table Construction

When determining the values of alternative elasticity parameters, such as the Armington parameter and CET parameter, the SAM table does not provide direct information. Therefore, other methods are used. Some researchers estimate elasticity using econometric approaches and historical data, while others rely on preset values based on previous studies or empirical estimations by other scholars. In this study, we adopt the latter approach, referring to the research findings of scholars like Zhang [23] and Zhai [24]. Table 1 presents substitution elasticity parameters for the production function, Table 2 displays Armington substitution elasticity parameters, and Table 3 shows CET (Constant Elasticity of Transformation) substitution elasticity parameters.

**Table 1.** Substitution elasticity parameters for the production function.

| Sectors | 01 | 02 | 03 | 04 | 05 | 06 | 07 | 08 | 09 | 10 | 11 |
|---|---|---|---|---|---|---|---|---|---|---|---|
| $\varepsilon_X$ | 0.3 | 0.3 | 0.3 | 0.3 | 0.3 | 0.3 | 0.3 | 0.3 | 0.3 | 0.3 | 0.3 |
| $\varepsilon_{KEL}$ | 0.8 | 0.8 | 0.8 | 0.8 | 0.8 | 0.8 | 0.8 | 0.8 | 0.8 | 0.8 | 0.8 |
| $\varepsilon_{KE}$ | 0.6 | 0.6 | 0.6 | 0.6 | 0.6 | 0.6 | 0.6 | 0.6 | 0.6 | 0.6 | 0.6 |
| $\varepsilon_E$ | 1.2 | 1.2 | 1.2 | 1.2 | 1.2 | 1.2 | 1.2 | 1.2 | 1.2 | 1.2 | 1.2 |
| $\varepsilon_{fos}$ | 1.2 | 1.2 | 1.2 | 1.2 | 1.2 | 1.2 | 1.2 | 1.2 | 1.2 | 1.2 | 1.2 |
| $\varepsilon_{pg}$ | 1.3 | 1.3 | 1.3 | 1.3 | 1.3 | 1.3 | 1.3 | 1.3 | 1.3 | 1.3 | 1.3 |
| $\varepsilon_{coal}$ | 1.25 | 1.25 | 1.25 | 1.25 | 1.25 | 1.25 | 1.25 | 1.25 | 1.25 | 1.25 | 1.25 |
| $\varepsilon_{petr}$ | 1.6 | 1.6 | 1.6 | 1.6 | 1.6 | 1.6 | 1.6 | 1.6 | 1.6 | 1.6 | 1.6 |
| $\varepsilon_{gas}$ | 1.25 | 1.25 | 1.25 | 1.25 | 1.25 | 1.25 | 1.25 | 1.25 | 1.25 | 1.25 | 1.25 |
| $\varepsilon_{pow}$ | 2 | 2 | 2 | 2 | 2 | 2 | 2 | 2 | 2 | 2 | 2 |
| **Sectors** | **12** | **13** | **14** | **15** | **16** | **17** | **18** | **19** | **20** | **21** | |
| $\varepsilon_X$ | 0.3 | 0.3 | 0.3 | 0.3 | 0.3 | 0.3 | 0.3 | 0.3 | 0.3 | 0.3 | |
| $\varepsilon_{KEL}$ | 0.8 | 0.8 | 0.8 | 0.8 | 0.8 | 0.8 | 0.8 | 0.8 | 0.8 | 0.8 | |
| $\varepsilon_{KE}$ | 0.6 | 0.6 | 0.6 | 0.6 | 0.6 | 0.6 | 0.6 | 0.6 | 0.6 | 0.6 | |
| $\varepsilon_E$ | 1.2 | 1.2 | 1.2 | 1.2 | 1.2 | 1.2 | 1.2 | 1.2 | 1.2 | 1.2 | |
| $\varepsilon_{fos}$ | 1.2 | 1.2 | 1.2 | 1.2 | 1.2 | 1.2 | 1.2 | 1.2 | 1.2 | 1.2 | |
| $\varepsilon_{pg}$ | 1.3 | 1.3 | 1.3 | 1.3 | 1.3 | 1.3 | 1.3 | 1.3 | 1.3 | 1.3 | |
| $\varepsilon_{coal}$ | 1.25 | 1.25 | 1.25 | 1.25 | 1.25 | 1.25 | 1.25 | 1.25 | 1.25 | 1.25 | |
| $\varepsilon_{petr}$ | 1.6 | 1.6 | 1.6 | 1.6 | 1.6 | 1.6 | 1.6 | 1.6 | 1.6 | 1.6 | |
| $\varepsilon_{gas}$ | 1.25 | 1.25 | 1.25 | 1.25 | 1.25 | 1.25 | 1.25 | 1.25 | 1.25 | 1.25 | |
| $\varepsilon_{pow}$ | 2 | 2 | 2 | 2 | 2 | 2 | 2 | 2 | 2 | 2 | |

Note: $\varepsilon_X$ represents the substitution elasticity parameters for factors and intermediate products, $\varepsilon_{KEL}$ for labor-capital-energy substitution elasticity, $\varepsilon_{KE}$ for capital-energy substitution elasticity, $\varepsilon_E$ for energy substitution elasticity, $\varepsilon_{fos}$ for fossil energy substitution elasticity, $\varepsilon_{pg}$ for oil and gas energy substitution elasticity, $\varepsilon_{coal}$ for coal energy substitution elasticity, $\varepsilon_{petr}$ for petroleum energy substitution elasticity, $\varepsilon_{gas}$ for natural gas substitution elasticity, while $\varepsilon_{pow}$ for electricity energy substitution elasticity.

**Table 2.** Substitution elasticity parameters for Armington.

| Categories | 01 | 02 | 03 | 04 | 05 | 06 | 07 | 08 | 09 | 10 | 11 |
|---|---|---|---|---|---|---|---|---|---|---|---|
| $\varepsilon_{QE}$ | 3 | 2.5 | 2.5 | 2.5 | 2.5 | 2.5 | 2.5 | 2.5 | 2.5 | 2.5 | 2 |
| **Categories** | **12** | **13** | **14** | **15** | **16** | **17** | **18** | **19** | **20** | **21** | |
| $\varepsilon_{QE}$ | 2.5 | 2.5 | 2.5 | 2.5 | 2.5 | 2.5 | 2.5 | 2.5 | 1.1 | 1.1 | |

Note: $\varepsilon_{QE}$ represents the Armington substitution elasticity parameters.

**Table 3.** Substitution elasticity parameters for CET.

| Categories | 01 | 02 | 03 | 04 | 05 | 06 | 07 | 08 | 09 | 10 | 11 |
|---|---|---|---|---|---|---|---|---|---|---|---|
| $\varepsilon_{QM}$ | 4 | 3.5 | 3.5 | 3.5 | 3.5 | 3.5 | 3.5 | 3.5 | 3.5 | 3.5 | 3 |
| **Categories** | **12** | **13** | **14** | **15** | **16** | **17** | **18** | **19** | **20** | **21** | |
| $\varepsilon_{QM}$ | 3 | 2.5 | 3.5 | 3.5 | 3.5 | 3.5 | 3.5 | 3.5 | 0.5 | 0.5 | |

Note: $\varepsilon_{QM}$ represents the CET (Constant Elasticity of Transformation) substitution elasticity parameters.

Furthermore, concerning the carbon emission coefficients for various energy types, as stated in the 'IPCC National Greenhouse Gas Inventories', the coefficients are as follows: coal emits 0.7476 tons of carbon per ton of standard coal, oil emits 0.5532 tons of carbon per ton of standard coal, and natural gas emits 0.4479 tons of carbon per ton of standard coal.

The macroeconomic closure condition is essential for achieving balance in the CGE model, allowing equations from various modules to form a solvable equation set and solve for endogenous variables. The equilibrium closure module mainly consists of savings and investment balance, international payment balance, and product market equilibrium. This paper adopts the neoclassical macroeconomic closure condition, which is the factor market equilibrium. According to the description of macroeconomic closure [25], there are three commonly used rules in domestic and foreign CGE models: neoclassical closure, Lewis closure rule, and Keynesian closure rule. (1) The neoclassical closure rule is based on neoclassical theory, in which investment and all prices, including factor prices and commodity prices, are endogenously determined by the model. The actual current supply of production factors, such as labor and capital, achieves full employment. (2) The Lewis closure rule is often used in studying economic problems of developing countries. In developing countries, an abundant labor market supply is common, and labor supply can be considered unlimited under certain labor prices, which is set as an exogenous variable in studies. However, there is a shortage of capital. (3) The Keynesian closure rule is based on the Keynesian theory, which suggests that macroeconomic downturns lead to surplus capital and idle labor. Therefore, the supply of labor and capital factors is sufficient, and factor prices can be exogenous. Demand for employment and capital is insufficient and can be set as endogenous.

In summary, this paper finds the neoclassical macroeconomic closure to be suitable for our purposes. It is difficult to set a realistic curve for unemployment rates and factor returns, making the direct assumption of full employment relatively objective. In China's current situation, there is complete competition in both labor and capital markets, resulting in no unemployment or surplus capital. The CGE model construction and SAM table are detailed in Appendix A.

## 4. Policy Simulation

The current energy pricing system in China involves the liberalization of primary energy prices, while secondary energy prices are still regulated to varying extents. This paper considers important factors affecting energy price fluctuations in China and analyzes them within policy scenarios. These include: (1) The continuous upward trend of fossil energy prices over the past 20 years, as indicated by the National Bureau of Statistics' mining industry price index (Figure 1). (2) The implementation of carbon trading, environmental

taxes, and the future introduction of a carbon tax, leading to increased costs associated with fossil energy use. (3) The reform of the electricity market, which impacts electricity prices and their transmission within the economic system. The Chinese government's 'Several Opinions on Further Deepening the Reform of the Electricity System' aims to promote the marketization of electricity trading. Taking into account previous research and the reality of energy price fluctuations in China, this paper constructs a CGE model that subdivides the energy sectors. It systematically analyzes the impact mechanisms of energy price fluctuations on China's energy-environment-economy system under different policy scenarios, such as rising fossil energy prices and the marketization reform of the electricity sector. The current coal-electricity conflict in China is caused by government regulation of electricity prices. This paper aims to simulate the impact of coal price fluctuations under electricity price regulation on the economy, energy, and environment. It also examines the impact of crude oil price fluctuations under refined oil price regulation and natural gas price fluctuations under gas price regulation. The ultimate goal is to achieve marketization through energy price reform. This paper analyzes the impact of marketization scenarios for different energy prices on the economy, energy, and environment. Specifically, it focuses on rising energy prices and studies the influence of primary energy price fluctuations on the national economy, considering market mechanisms and government regulation of secondary energy prices while assuming other variables remain constant.

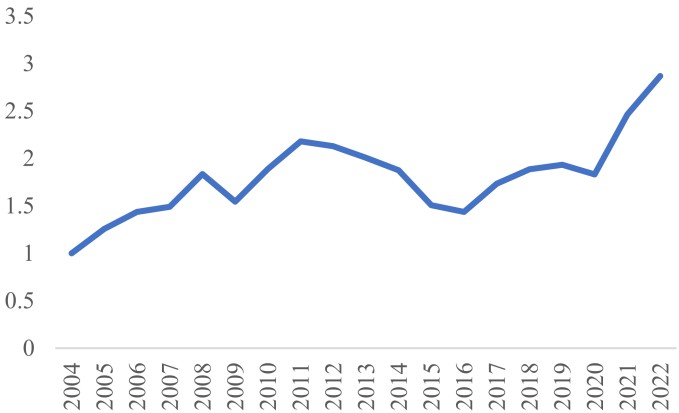

**Figure 1.** Producer price index for extractive industries.

### 4.1. Impact of Coal Price Fluctuations on the National Economy

Coal remains the primary energy source in China and a crucial aspect of the country's energy structure, given its natural resource endowment. As such, comprehending the influence of coal price swings on the national economy is of paramount importance from the perspective of national energy security. Coal accounts for approximately half of China's total energy consumption and is critical to ensuring electricity supply. In contrast to other energy sources, coal prices in China have been mostly liberalized. While the country is actively pursuing marketization reforms in the electricity sector, electricity prices remain subject to government regulation. This study focuses on examining the impact of coal price fluctuations on the national economy by analyzing thermal power price market mechanisms and government regulation. Thermal power constitutes the most significant direct downstream industry of the coal industry, and thus, it is imperative to study its dynamics to provide a comprehensive understanding of the subject matter.

#### 4.1.1. Sectoral Prices, Output, and Investment Consumption

As shown in Figure 2, as an upstream industry in the industrial chain, an increase in coal prices leads to varying degrees of price increases in various sectors. Under the market mechanism of thermal power prices, the coal-electricity industry chain smoothly transmits the impact. With a 5% increase in coal prices, thermal power prices rise by 1.56%, which then affects downstream industries such as mining, machinery equipment manufacturing,

and others. The coal-coking industry chain exhibits similar effects, with the coking industry and metal smelting industry prices rising by 2.40% and 0.59%, respectively. The coal chemical industry chain also has a significant impact, resulting in a 0.39% price increase in the chemical industry, affecting downstream sectors such as textiles and their products. Lastly, the coal-building materials industry chain is worth noting, as the non-metallic mineral products industry prices rise by 0.73%, followed by an increase in construction industry prices. Other sectors such as agriculture, service industry, and food industry, which are downstream in the industrial chain, are less affected by the rise in coal prices. The increase in coal prices leads to an increased demand for alternative energy sources, causing petroleum and natural gas prices to rise to varying degrees.

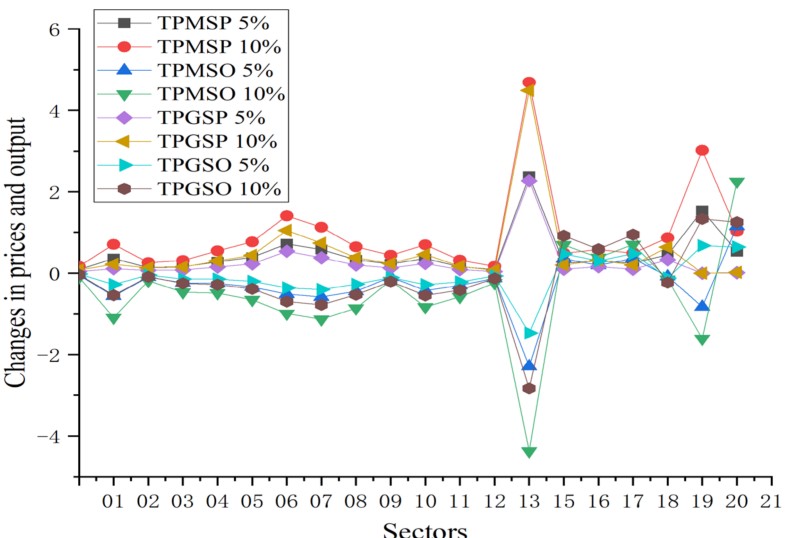

**Figure 2.** Changes in price and output due to coal price volatility (TPM is thermal power price marketization, TPG is thermal power price regulation, SP is price change, SO is output change).

Figure 2 demonstrates that government regulation of electricity prices disrupts the normal price transmission within the coal-electricity supply chain, lessening the impact of coal price fluctuations on downstream industries. As a result, compared to a market-driven approach to thermal power pricing, the decrease in industrial sector prices is less significant. Industries such as chemicals, coking, and non-metallic mineral products, which have high electricity and direct coal demands, experience a minor impact on their prices due to regulated thermal power prices. For example, a 5% increase in coal prices results in price rises of 2.29%, 0.10%, 0.55%, 0.38%, and 0.23% in these sectors, respectively. Government intervention in electricity pricing also reduces the substitution effect between different energy sources, leading to a smaller increase in petroleum and natural gas prices under this regime. Furthermore, the output across various sectors declines less under government-regulated thermal power prices than in a market-based scenario. Specifically, the coking, chemical, and metal smelting industries see notable output declines, while sectors like mining, textiles, and machinery equipment manufacturing are more resilient to government price controls. By lowering energy input costs for companies, government regulation helps mitigate the reduction in sector outputs compared to market-driven thermal power prices.

### 4.1.2. Sectoral Energy Consumption and Carbon Emissions

Figure 3 shows that under the market mechanism of thermal power prices, increasing coal prices leads to a significant decrease in energy consumption in high-energy-consuming industrial sectors. For instance, with a 5% increase in coal prices, the coking industry's energy demand decreases by 3.07%, while thermal power decreases by 1.29%, and non-metallic mineral products and metal smelting industries decrease by around 1.52%, with other industrial sectors also experiencing noticeable declines. Although petroleum pro-

cessing increases its energy consumption due to substitution effects, the increase is not significant, indicating that the rise in coal prices plays a vital role in curbing China's energy consumption. However, government regulation of electricity prices weakens the effect of coal price increases in suppressing energy consumption, except for some industries directly downstream of coal, such as coking, chemical, and metal smelting industries, which experience relatively smaller decreases. The energy consumption of other industrial sectors decreases by more than 50% compared to the situation under the market mechanism of thermal power prices, as electricity appears cheaper and enterprises have less incentive to advance technology, resulting in a smaller decrease in energy intensity. Although regulating electricity prices ensures economic growth, it has negative impacts on reducing carbon emissions and improving energy efficiency.

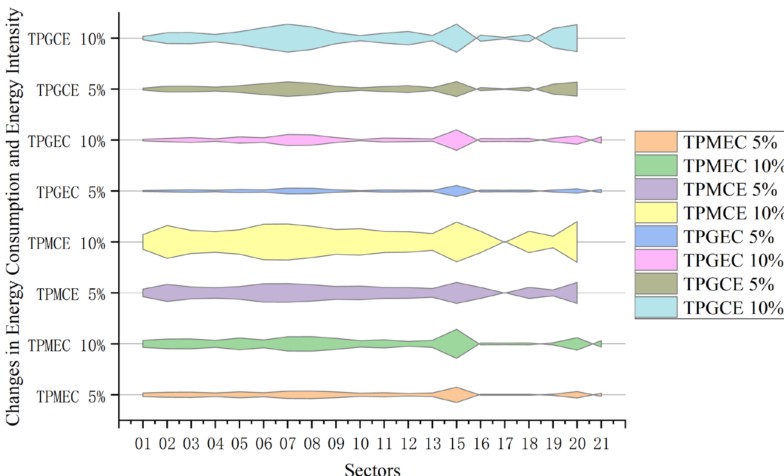

**Figure 3.** Changes in energy consumption and carbon emissions due to coal price volatility (TPM is thermal power price marketization, TPG is thermal power price regulation, EC is change in energy consumption, CE is change in carbon emissions).

Figure 3 demonstrates that a rise in coal prices generally leads to reduced coal demand across most industrial sectors. Since coal is a major source of $CO_2$ emissions, this price increase significantly cuts carbon emissions industry-wide. In a market-driven thermal power pricing system, sectors with high energy consumption exhibit the largest drops in carbon emissions. Specifically, a 5% hike in coal prices results in the most substantial emission reductions in thermal power, coking, metal smelting, and non-metallic mineral processing industries, with decreases of 4.23%, 4.19%, 3.26%, and 3.75%, respectively. Other sectors also see varied levels of reduction. Despite increased energy consumption from petroleum and natural gas due to substitution effects, their reduced coal demand—and coal's high carbon emission factor—mean these sectors also see a decrease in emissions. However, government regulation of electricity prices disrupts the coal-electricity supply chain's price signals, significantly dampening the impact of coal price rises on reducing carbon emissions. While emissions still fall under government-controlled electricity prices, the reduction is much less pronounced than with market mechanisms. The chemical and metal smelting industries, despite being influenced, exhibit more substantial emission declines under government regulation, with carbon emissions dropping by 2.19% and 2.34%, respectively, following a 5% increase in coal prices.

### 4.1.3. Macroeconomic Variables and Resident Welfare

As depicted in Figure 4, the thermal power pricing mechanism reveals that a surge in coal prices adversely affects the incomes of residents, businesses, and the government, with businesses bearing the brunt. Specifically, a 5% hike in coal prices results in a 0.08% drop in business income, while government and resident incomes decline by 0.05% and 0.009%, respectively. This price increase also triggers a contraction in the macroeconomy,

decreasing investment, savings, and trade (imports/exports) by 0.07%, 0.13%, and 0.12%, respectively, which in turn leads to a reduction in real GDP. As coal prices escalate, inflation follows, causing the price index to rise and thereby mitigating the fall in nominal GDP compared to real GDP. Furthermore, the uptick in the price index coupled with diminished resident income curtails resident welfare, evidenced by a 227.41 decrease following a 5% rise in coal prices. However, under a regulated electricity pricing scenario, the transmission of coal price hikes to general product prices is interrupted, softening their impact on macroeconomic indicators. In both scenarios, a 5% coal price increase reduces enterprise income by 0.08% and 0.05%, but the effect on government income is more pronounced, dropping by 0.05% and 0.02%. The downturn in other macroeconomic figures is also notably lessened, with real and nominal GDP witnessing declines of 0.13% and 0.02%, respectively, and resident welfare diminishing by 102.09.

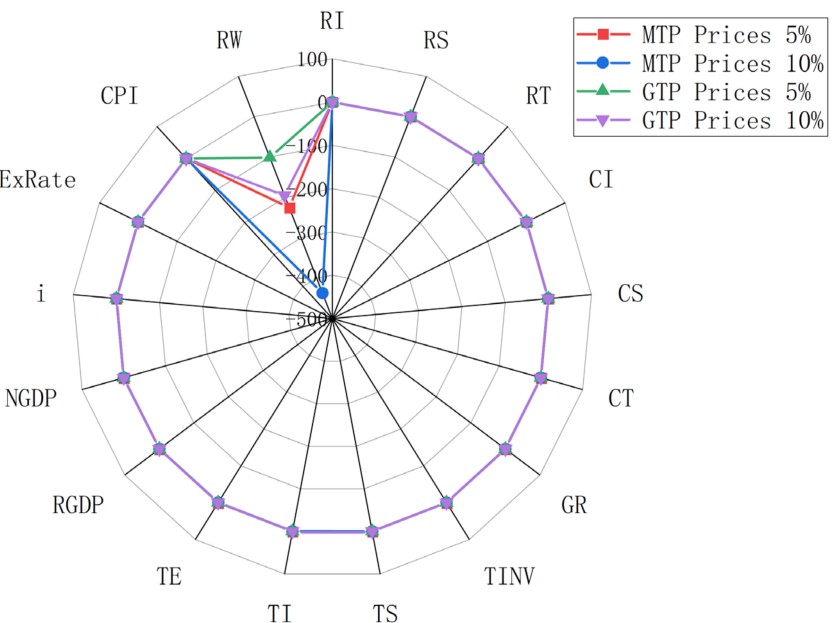

**Figure 4.** Impact of coal price volatility on the macroeconomy and welfare of the population (MTP is the Market Mechanism for Thermal Power Prices, GTP is the Market Regulation of Thermal Power Prices).

### *4.2. Impact of Oil Price Fluctuations on the National Economy*

Petroleum, a crucial energy source in China, is pivotal for economic growth and national security. Since 1998, the Chinese government has embarked on market reforms within the petroleum sector. The crude oil market has largely achieved marketization, accompanied by five rounds of reforms in the refined oil market, indirectly tethering it to the international market. Nonetheless, while a minor portion of refined oil prices has been liberalized, the majority remains under government control to varying extents. Given that refined oil is the most immediate downstream product of crude oil, this paper examines the dynamics from two angles: the market mechanism of refined oil prices and the extent of government regulation.

### 4.2.1. Sectoral Prices, Output, and Investment Consumption

As illustrated in Figure 5, a surge in crude oil prices triggers varying levels of price hikes across different industrial sectors, yet its impact is more sector-specific compared to coal. The influence of crude oil on diverse industrial sectors primarily flows through two industry chains. The first is the crude oil-to-refined oil chain. Within the refined oil pricing market mechanism, an uptick in crude oil prices directly escalates refined oil prices. A 5% increase in crude oil prices leads to a 1.62% rise in refined oil prices, subsequently inflating costs in the transportation, electricity, and other industrial sectors, with the transportation

sector being notably sensitive to crude oil price fluctuations, experiencing a 0.34% price increase. Additionally, the refining process of crude oil produces some gas, meaning that crude oil price hikes indirectly boost gas prices as well. The second pathway is the crude oil-to-chemical industry chain. Given that crude oil is a vital raw material for the materials industry, including steel and fertilizers, its price hike propels increases in the chemical, coking, and agricultural sectors, with a 5% rise in crude oil prices inducing respective increases of 0.28%, 0.20%, and 0.06% in these sectors.

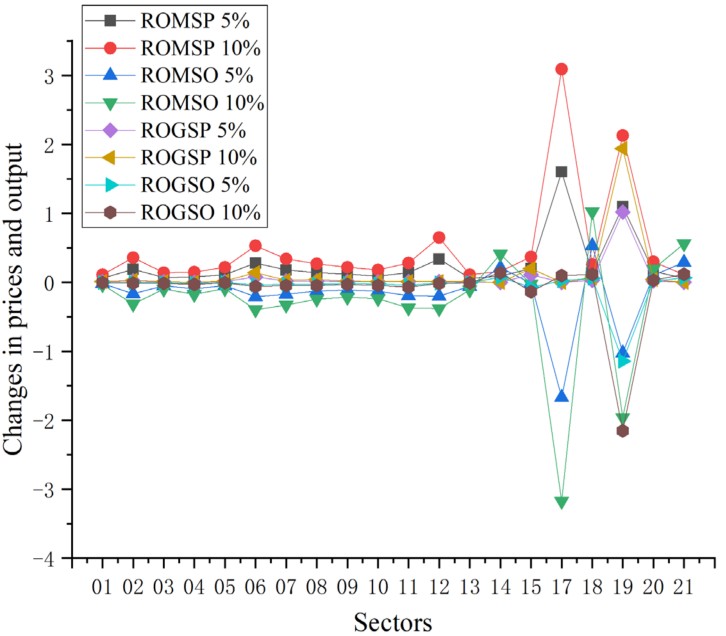

**Figure 5.** Changes in prices and output due to oil price volatility (ROM is marketization of refined oil prices, ROG is refined oil price regulation, SP is price change, and SO is output change).

As depicted in Figure 5, government regulation of refined oil prices significantly mitigates the impact of rising crude oil prices on various industrial sectors, leaving some sectors entirely unaffected by crude oil price fluctuations. Given that most industrial sectors rely on refined rather than crude oil, government control over refined oil prices renders the effect of crude oil price increases on these sectors negligible. With the crude oil-to-refined oil industry chain effectively obstructed, the influence of the crude oil-to-chemical industry chain becomes more pronounced during crude oil price surges. A 5% hike in crude oil prices results in just a 0.14% increase in chemical industry prices. Under government regulation, the transmission of crude oil price changes to downstream sectors is almost entirely severed, exerting little to no impact on the output across various sectors. Aside from some direct downstream and energy-intensive sectors like gas production, coking, and chemical industries, the output fluctuations in other industrial sectors are minimal, with most experiencing less than a 0.1% decline in output.

### 4.2.2. Sectoral Energy Consumption and Carbon Emissions

As illustrated in Figure 6, rising crude oil prices result in a decrease in energy consumption across various sectors, primarily affecting petroleum use. Given the characteristics of China's energy consumption structure, the reduction in sector-wide energy consumption due to increased crude oil prices is less pronounced than that for coal. Moreover, due to the differing elasticities of energy demand among industrial sectors, the impact of rising crude oil prices on energy consumption varies. The petroleum processing and chemical industries experience significant decreases in energy consumption, with a 5% increase in crude oil prices leading to a 1.86% and 0.58% reduction in energy demand, respectively. The transportation sector, as the largest consumer of refined oil, witnesses a 0.83% drop

in energy demand following a 5% hike in crude oil prices. Conversely, coal and natural gas consumption see an uptick due to substitution effects. Other industrial sectors also curtail their energy consumption in response to rising crude oil prices. Under government regulation of refined oil prices, the transmission of crude oil price increases across industrial sectors is minimal, leaving the energy demand of various sectors largely unaffected by rising crude oil prices. The chemical industry, a key downstream sector of crude oil, exhibits a notable decrease in energy demand, with a 5% increase in crude oil prices resulting in a 0.17% reduction. Gas consumption, another important downstream sector, experiences a 1.48% decline in energy use, showing a lesser impact of government regulation on refined oil prices. Other industrial sectors display minor adjustments, with a 5% rise in crude oil prices causing less than a 0.10% decrease in energy demand across these sectors.

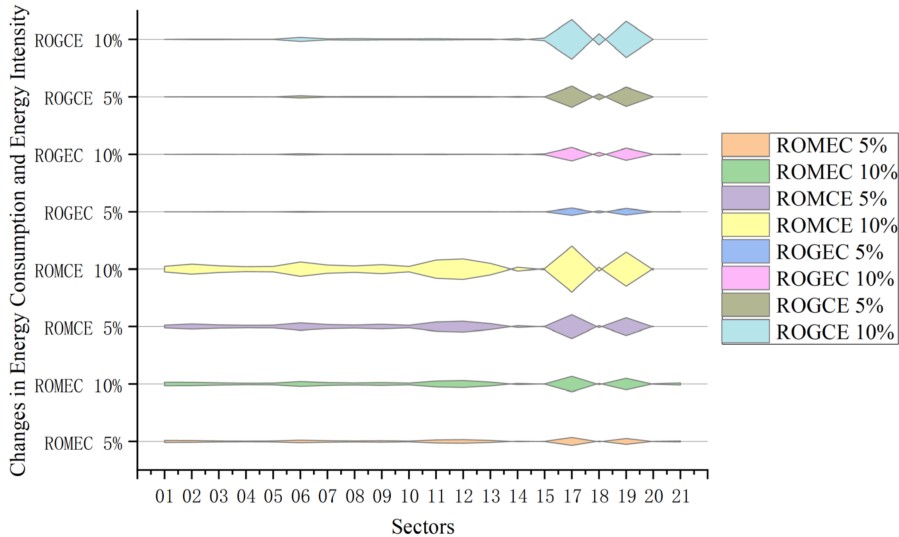

**Figure 6.** Changes in energy consumption and carbon emissions as a result of oil price volatility (ROM is the marketization of refined oil prices, ROG is the regulation of refined oil prices, EC is the change in energy consumption, and CE is the change in carbon emissions).

As depicted in Figure 6, within the context of refined oil price market mechanisms, crude oil's smaller share in China's energy mix compared to coal, along with its lower carbon emission coefficient, results in a less pronounced decrease in sector-wide carbon emissions following a rise in crude oil prices than that observed with coal. The increase in crude oil prices notably affects carbon emissions in high-energy-consuming industries. A 5% hike in crude oil prices leads to a significant 5.50% reduction in carbon emissions in the petroleum processing industry. In parallel, carbon emissions in the transportation sector, gas production industry, and metal smelting industry decrease by 2.46%, 4.10%, and 0.73%, respectively. Furthermore, a 5% increase in crude oil prices results in a 1.71% decline in carbon emissions in the chemical industry, while the textile product industry and agriculture, both heavily reliant on chemical products, see carbon emissions decrease by 0.61% and 0.64%, respectively. Under government regulation of refined oil prices, the transmission of crude oil price increases to downstream industrial sectors is effectively halted, leading to a more modest reduction in carbon emissions. Sectors with a direct reliance on crude oil, such as the chemical industry, exhibit notable shifts in carbon emissions, with a 5% increase in crude oil prices causing a 0.39% reduction in the chemical industry's carbon emissions. Other high-energy-consuming sectors also experience more substantial drops in carbon emissions, with the mining, non-metallic mineral processing, and metal smelting industries seeing decreases of 0.23%, 0.38%, and 0.39%, respectively.

### 4.2.3. Macroeconomic Variables and Resident Welfare

Figure 7 reveals that, similar to coal, a surge in crude oil prices negatively affects the income of households, businesses, and the government within the industrial chain, albeit to a lesser extent than coal. A 5% increase in crude oil prices under the market dynamics of refined oil prices results in marginal income reductions of 0.0006% for households, 0.06% for businesses, and 0.05% for the government. This price hike also impacts other macroeconomic indicators, leading to a 0.12% and 0.03% drop in real and nominal GDP, respectively, a 0.06% decrease in interest rates, and a 0.16% increase in exchange rates. Consequently, resident welfare declines by 144.66 due to the combined effect of rising price indexes and falling incomes. However, Figure 7 also illustrates that government regulation of refined oil prices significantly mitigates the economic downturn caused by rising crude oil prices. The effects on imports, exports, and real GDP are particularly softened, with a 5% increase in crude oil prices causing just a 0.0023% and 0.0016% decrease in imports and exports, respectively. Notably, government intervention reduces the decline in real GDP by 80% compared to the market-driven scenario, highlighting the importance of price controls in sustaining economic growth. Furthermore, the impact on resident welfare is considerably less severe under government regulation, with a 5% and 10% increase in crude oil prices resulting in welfare reductions of only 21.47 and 35.49, respectively.

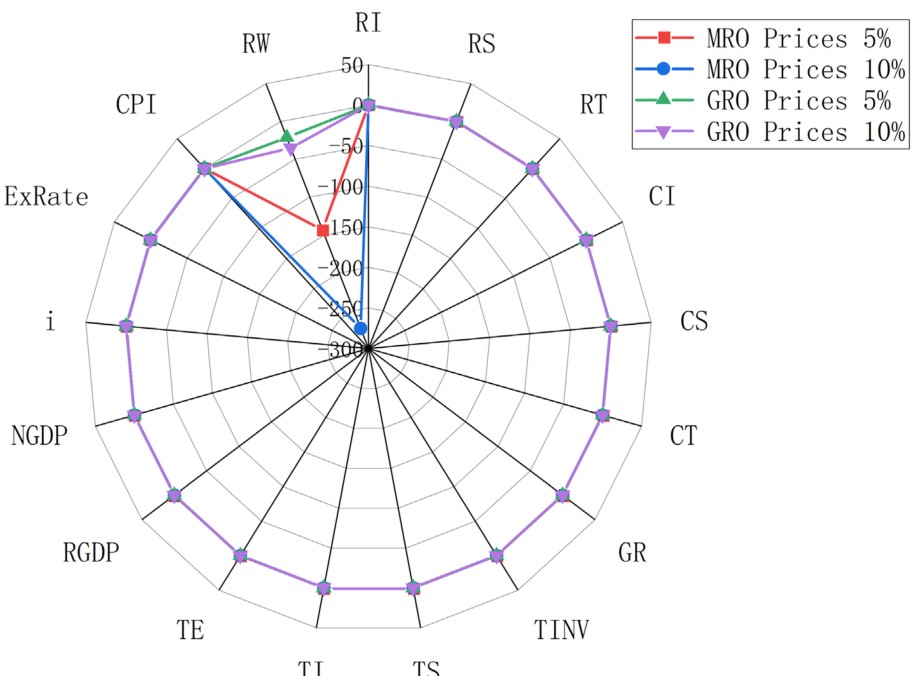

**Figure 7.** Impact of oil price volatility on the macroeconomy and welfare of the population (MRO is the Market Mechanism for Oil Product Prices and GTP is the Market Regulation of Oil Product Prices).

### 4.3. Impact of Natural Gas Price Fluctuations on the National Economy

The consensus on vigorously promoting natural gas as a strategy to further reduce pollutant emissions is clear. Yet, the current pricing mechanism for natural gas fails to accurately mirror market fluctuations and the intrinsic value of the resource, hindering its exploration and development. Consequently, reforming natural gas prices is crucial for aligning its cost with that of other energy sources more logically. This paper examines the impact of natural gas pricing from two angles: the market-driven pricing mechanism and government-regulated pricing, drawing parallels with the downstream industries of coal and crude oil.

### 4.3.1. Sectoral Prices, Output, and Investment Consumption

Figure 8 illustrates that currently, natural gas holds a smaller share in China's energy mix and exerts a less significant influence on various sectors compared to coal and oil. As a key component of the industrial chain, an increase in natural gas prices impacts downstream industries, leading to varying degrees of price hikes across different sectors. The downstream natural gas industry encompasses transportation, utility (gas), and chemical sectors. With the rise in oil prices and concerns over air quality, an increasing number of vehicles, including city buses and taxis, are switching to natural gas, making the transportation sector more sensitive to changes in natural gas prices. A 5% hike in natural gas prices results in a 0.15% increase in transportation costs. Similarly, the chemical industry, which heavily relies on natural gas as a feedstock, faces notable price escalations with a 5% increase in natural gas prices causing price rises of 0.12% and 0.70% in the chemical and petroleum processing sectors, respectively.

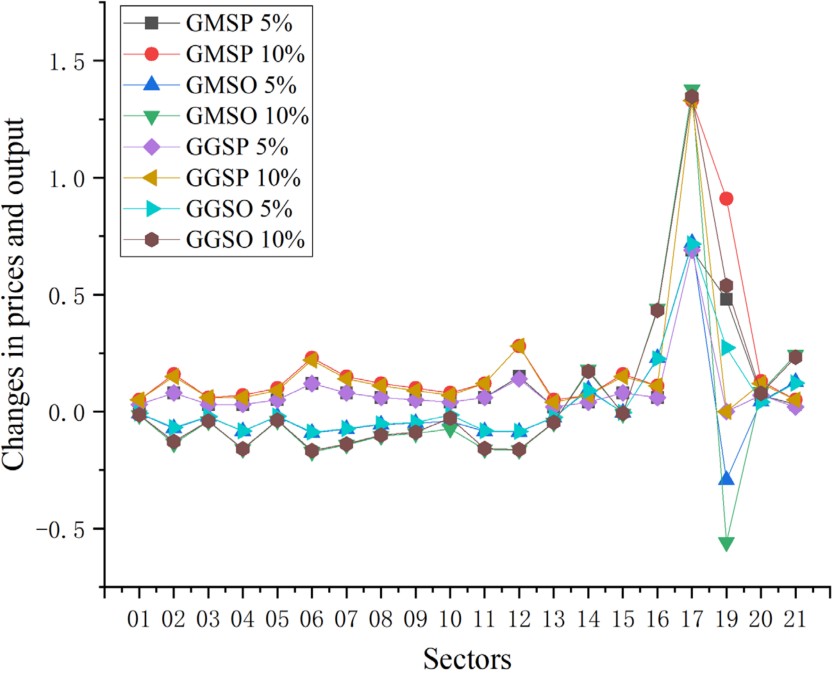

**Figure 8.** Changes in price and output as a result of gas price volatility (GM is gas price marketization, GG is gas price regulation, SP is price change, and SO is output change).

Figure 8 demonstrates that government regulation of gas prices impedes the flow from the natural gas industry to downstream sectors, thereby mitigating the effects of natural gas price hikes on gas-related industries. However, given that gas is just one segment of the broader natural gas downstream industry and represents a minor share of the overall energy consumption structure, government control over gas prices does not markedly influence the transmission of natural gas price changes to other industrial chains. Significant sectoral price adjustments only emerge when natural gas prices increase by 10%. Since gas constitutes less than one-third of the natural gas downstream and its sources have diversified in recent years, government-regulated gas prices minimally impact sector output and consumer consumption. For instance, in the transportation sector, a 5% increase in natural gas prices under government regulation leads to a mere 0.08% reduction in output, a negligible difference of just 0.01 percentage points compared to scenarios governed by market-driven gas prices.

### 4.3.2. Sectoral Energy Consumption and Carbon Emissions

Figure 9 clearly shows that the effect of rising natural gas prices on energy consumption differs across sectors, primarily driven by a decline in natural gas demand. In a

market-based gas pricing scenario, an increase in natural gas prices leads notably to reduced energy consumption in sectors such as gas production, the chemical industry, and transportation, with decreases of 0.45%, 0.25%, and 0.36%, respectively following a 5% price hike. Traditional sectors with high energy use, like mining and construction, also see significant cuts in energy usage, declining by 0.17% and 0.32%, respectively. Additionally, the agricultural sector feels the impact, with a 0.19% drop in energy demand due to higher natural gas prices. Under government-regulated gas prices, the pattern of reduction in energy consumption across different sectors persists, albeit to a lesser extent. With a 5% rise in natural gas prices, the decreases in energy consumption for gas production, the chemical industry, and transportation are slightly lower at 0.38%, 0.25%, and 0.36%, respectively. This indicates that while government control of gas prices does mitigate the overall impact on energy consumption, the effect remains noticeable in specific industries.

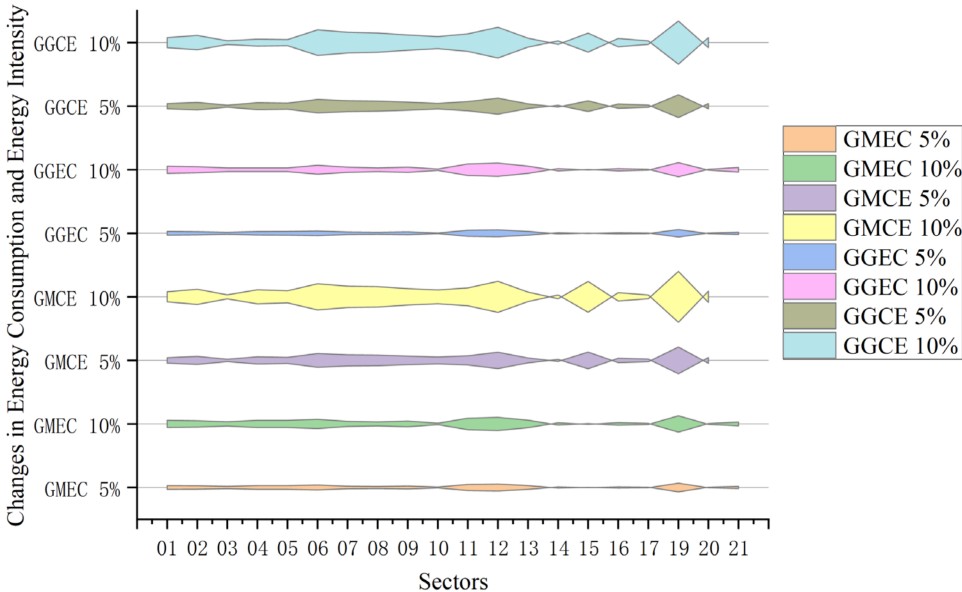

**Figure 9.** Impact of gas price volatility on energy consumption and carbon emissions (GM is gas price marketization, GG is gas price regulation, EC is change in energy consumption, CE is change in carbon emissions).

Figure 9 reveals that the rise in natural gas prices has a negligible effect on the energy consumption of various industrial sectors. Given that natural gas emits less carbon than coal and crude oil, its impact on sector-wide carbon emissions is considerably smaller. In a market-driven pricing environment, the increase in natural gas prices primarily affects carbon emissions in high-energy-consuming industries such as chemicals, transportation, and coking. A 5% hike in natural gas prices results in carbon emission reductions of 0.72%, 0.85%, and 0.87% in these sectors, respectively, with other industries experiencing minor declines. Under government-regulated gas prices, the slight change in energy consumption across sectors mirrors the market scenario, leading to similarly modest impacts on carbon emissions. For instance, a 5% rise in government-controlled gas prices decreases carbon emissions by 0.72%, 0.85%, and 0.56% in the chemical, transportation, and coking sectors, respectively, showing little deviation from market-driven outcomes.

### 4.3.3. Macroeconomic Variables and Household Welfare

Figure 10 clearly shows that among all fossil fuels, natural gas has the least impact on macroeconomic indicators and household welfare due to its minimal share in primary energy sources. A 5% increase in natural gas prices under a market-based pricing mechanism results in an almost negligible 0.0003% drop in household income. Similarly, corporate income and government revenue see minor decreases of 0.026% and 0.009%, respectively. Other impacts include a 0.02% fall in total investment, a 0.07% reduction in total imports, a

0.06% decrease in exports, a 0.05% decrease in real GDP, a slight 0.01% dip in nominal GDP, a 0.03% reduction in interest rates, a 0.07 increase in exchange rates, a 0.04% rise in the consumer price index, and a 62.80 decrease in household welfare. The figure also indicates that government regulation of gas prices does little to buffer the national economy from the effects of rising natural gas prices, with the impact remaining marginal. Key variables such as household income, interest rates, exchange rates, and the consumer price index show no significant change under a 5% hike in controlled gas prices, mirroring the market mechanism scenario. Other macroeconomic indicators, including real and nominal GDP, exhibit minimal changes, decreasing by 0.05% and 0.01%, respectively, with household welfare decreasing by 56.82, highlighting the limited influence of natural gas price fluctuations on the broader economy.

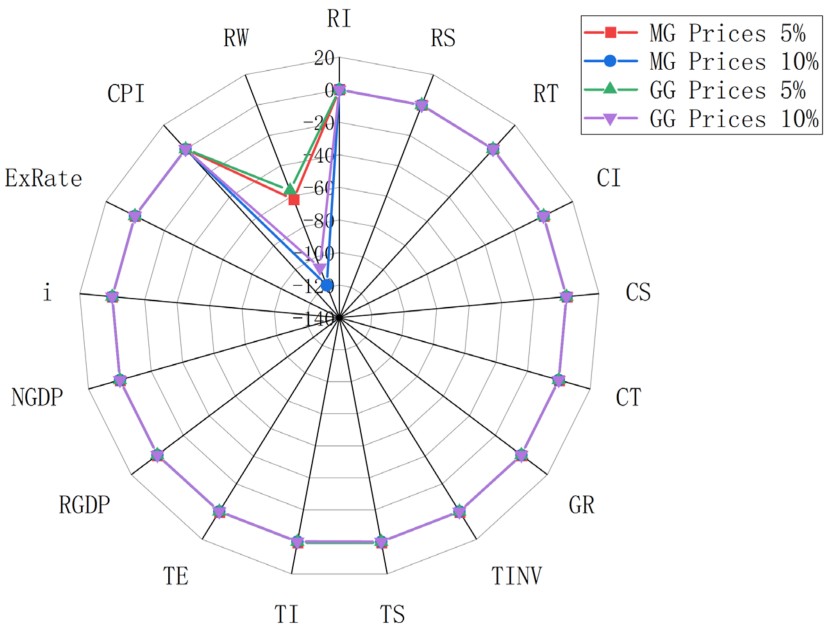

**Figure 10.** Impact of gas price volatility on the macroeconomy and welfare of the population (MG for gas price market mechanism, GG for gas price market regulation).

### 4.4. Impact of Refined Oil and Natural Gas Price Fluctuations on the National Economy

Given that downstream industrial sectors and consumers primarily rely on secondary energy sources, including refined oil, the pricing of these energies often falls under varying degrees of government oversight. Energy price reform is largely centered around adjusting the prices of secondary energies like refined oil. With China's economic expansion, the demand for refined oil continues to rise. Therefore, streamlining the pricing system for refined oil to more accurately reflect market demand is crucial for reducing government subsidies and fiscal pressures, conserving energy resources, ensuring national energy security, and bolstering the capacity of energy services to support socio-economic growth. Additionally, revising the natural gas pricing mechanism is key to setting reasonable energy price ratios, encouraging the exploration and development of upstream natural gas resources, and increasing the share of natural gas in the energy mix. This chapter delves into the mechanisms of the refined oil and natural gas markets from these perspectives.

### 4.4.1. Sectoral Price Output and Investment Consumption

Table 4 illustrates that refined oil, due to its broad applications and the necessity for most crude oil to be processed into refined oil for use, has a more pronounced impact on the national economy than crude oil price increases. Refined oil's influence on downstream industrial sectors manifests through two primary channels. The first channel involves the refined oil-chemical industry link, where rising refined oil prices escalate costs within the chemical sector, subsequently affecting industries such as construction and paper printing.

Additionally, sectors like metal smelting and electricity are part of refined oil's downstream impact, leading to notable price increases in the chemical industry, other mining sectors, and the non-metallic mineral products industry. Specifically, a 5% hike in refined oil prices results in price jumps of 0.62%, 0.55%, and 0.52% in these sectors, respectively. The second channel is the refined oil-transportation industry connection. As a crucial fuel source for transportation, refined oil price surges exert the most significant effect on this sector. Among all industries, the transportation sector experiences the highest price escalation, with a 5% rise in refined oil prices causing a 1.06% increase in prices.

**Table 4.** The influence of oil and gas price fluctuations on the department's price and output (%).

| | Refined Oil Price Market Mechanism | | | | Gas Price Market Mechanism | | | |
|---|---|---|---|---|---|---|---|---|
| | Sector Price | | Sector Output | | Sector Price | | Sector Output | |
| | 5% | 10% | 5% | 10% | 5% | 10% | 5% | 10% |
| Agriculture, Forestry, Animal Husbandry, and Fishery | 0.1717 | 0.3232 | −0.1272 | −0.2491 | 0.0101 | 0.0101 | −0.0042 | −0.0082 |
| Other Mining Industries | 0.5555 | 1.0807 | −0.5354 | −1.0431 | 0.0404 | 0.0808 | −0.0452 | −0.0877 |
| Food and Tobacco Industry | 0.2222 | 0.4343 | −0.1814 | −0.3538 | 0.0101 | 0.0202 | −0.0088 | −0.0171 |
| Textile Products Industry | 0.2525 | 0.4949 | −0.1570 | −0.3002 | 0.0101 | 0.0303 | −0.0079 | −0.0155 |
| Wood and Paper Industry | 0.3333 | 0.6464 | −0.2306 | −0.4516 | 0.0202 | 0.0303 | −0.0115 | −0.0222 |
| Chemical Industry | 0.6464 | 1.2625 | −0.5518 | −1.0711 | 0.0303 | 0.0707 | −0.0287 | −0.0554 |
| Non-Metal Mineral Industry | 0.5151 | 1.0100 | −0.4663 | −0.9075 | 0.0202 | 0.0505 | −0.0212 | −0.0410 |
| Metal Smelting Industry | 0.3838 | 0.7575 | −0.3239 | −0.6321 | 0.0303 | 0.0606 | −0.0261 | −0.0506 |
| Machinery and Equipment Industry | 0.3333 | 0.6565 | −0.3031 | −0.5915 | 0.0202 | 0.0505 | −0.0250 | −0.0485 |
| Communications, Instruments Industry | 0.2727 | 0.5353 | −0.3241 | −0.6238 | 0.0202 | 0.0404 | −0.0015 | −0.0031 |
| Construction Industry | 0.4343 | 0.8484 | −0.4993 | −0.9714 | 0.0202 | 0.0303 | −0.0204 | −0.0395 |
| Transportation and Postal Industry | 1.0605 | 2.0806 | −0.5965 | −1.1572 | 0.0101 | 0.0303 | −0.0176 | −0.0338 |
| Service Industry | 0.1919 | 0.3737 | −0.1692 | −0.3297 | 0.0101 | 0.0202 | −0.0101 | −0.0195 |
| Coal Mining and Washing Industry | 0.2727 | 0.5353 | 0.4513 | 0.8746 | 0.0101 | 0.0202 | 0.0411 | 0.0795 |
| Coking Industry | 0.2929 | 0.5757 | −0.2562 | −0.4943 | 0.0707 | 0.1414 | −0.0553 | −0.1072 |
| Petroleum Extraction Industry | 0.3434 | 0.6565 | −0.1917 | −0.3469 | 0.0202 | 0.0303 | 0.0429 | 0.0817 |
| Petroleum Processing Industry | 5.0500 | 10.1000 | −5.3204 | −10.0924 | 0.0202 | 0.0303 | 0.0412 | 0.0800 |
| Natural Gas Extraction Industry | 0.3434 | 0.6565 | 1.1549 | 2.2761 | 0.0202 | 0.0303 | −0.1384 | −0.2672 |
| Gas Production Industry | 0.3636 | 0.7171 | 0.2539 | 0.4957 | 5.0500 | 10.1000 | −5.7785 | −10.9848 |
| Thermal Power | 0.4242 | 0.8282 | 0.2708 | 0.5252 | 0.0202 | 0.0404 | 0.0341 | 0.0664 |
| Clean Energy | 0.1919 | 0.3737 | 0.7319 | 1.4248 | 0.0101 | 0.0202 | 0.0548 | 0.1063 |

Table 4 shows that natural gas occupies a smaller fraction of overall energy consumption, leading to its price increases having a relatively modest effect on most industrial sectors. The sectors most sensitive to natural gas price fluctuations include coking, other mining industries, the chemical industry, and the non-metallic mineral products industry. A 5% hike in natural gas prices results in price increases of 0.07%, 0.04%, 0.03%, and 0.03% in these sectors, respectively. Compared to refined oil, natural gas constitutes a significantly lesser portion of the end-use energy mix. Despite its position upstream in the industrial chain, a rise in natural gas prices does not significantly affect sectoral outputs. The coking industry experiences a slightly more pronounced impact, with a 5% increase in natural gas prices leading to a 0.1% output reduction. Given its benefits, an increasing number of households are turning to natural gas for daily energy needs, making the service industry

particularly susceptible to natural gas price hikes. With natural gas prices rising by 5% and 10%, the service industry sees output reductions of 0.02% and 0.026%, respectively, highlighting the significant influence of natural gas pricing on this sector.

### 4.4.2. Sectoral Energy Consumption and Carbon Emissions

Table 5 reveals that a surge in refined oil prices results in a downturn in output across various industrial sectors, subsequently reducing refined oil consumption. This trend is notably pronounced in sectors like transportation, the chemical industry, and construction, which significantly cut back on their energy use. Specifically, a 5% hike in refined oil prices triggers a decrease in energy demand by 2.47%, 1.29%, and 2.09% in these sectors, respectively. As China's agriculture sector continues to mechanize and the usage of chemical fertilizers escalates, the impact of rising refined oil prices on this sector is considerable, leading to a marked drop in energy demand. Increases of 5% and 10% in refined oil prices correspond to reductions of 1.43% and 2.76% in energy demand within the agriculture, forestry, animal husbandry, and fishery sectors. Other industrial sectors also report varied levels of energy input declines due to the uptick in refined oil prices. Similarly, climbing gas prices curtail energy inputs across different industrial sectors, albeit with a lesser impact compared to refined oil. The effect is primarily observed in sectors with high gas consumption, such as the chemical industry, other mining industries, and the coking industry, where a 5% rise in gas prices leads to energy demand decreases of 0.08%, 0.11%, and 0.08%, respectively. Given that gas is a critical utility for daily life, its price increase exerts a significant influence on energy demand in the service industry, with a 5% increase in gas prices causing a 0.09% reduction in energy demand, a more substantial decline than seen in some industrial sectors.

**Table 5.** The influence of the oil and gas price fluctuation on the department's energy inputs (%).

| | Refined Oil Price Market Mechanism | | | | Gas Price Market Mechanism | | | |
|---|---|---|---|---|---|---|---|---|
| | Energy Consumption | | Carbon Emission | | Energy Consumption | | Carbon Emission | |
| | 5% | 10% | 5% | 10% | 5% | 10% | 5% | 10% |
| Agriculture, Forestry, Animal Husbandry, and Fishery | −1.4299 | −2.7605 | −2.0227 | −3.9052 | −0.0207 | −0.0401 | −0.1033 | −0.2001 |
| Other Mining Industries | −1.1879 | −2.2980 | −1.7543 | −3.3937 | −0.1133 | −0.2198 | −0.5655 | −1.0968 |
| Food and Tobacco Industry | −0.7473 | −1.4469 | −1.1036 | −2.1369 | −0.0465 | −0.0861 | −0.2319 | −0.4295 |
| Textile Products Industry | −0.4456 | −0.8670 | −0.6581 | −1.2805 | −0.0574 | −0.1112 | −0.2862 | −0.5549 |
| Wood and Paper Industry | −0.6933 | −1.3451 | −1.0238 | −1.9865 | −0.0419 | −0.0812 | −0.2092 | −0.4052 |
| Chemical Industry | −1.2939 | −2.4974 | −1.9108 | −3.6881 | −0.0724 | −0.1399 | −0.3614 | −0.6980 |
| Non-Metal Mineral Industry | −0.8818 | −1.7077 | −1.3023 | −2.5220 | −0.0386 | −0.0746 | −0.1925 | −0.3725 |
| Metal Smelting Industry | −0.5846 | −1.1354 | −0.8633 | −1.6769 | −0.0581 | −0.1125 | −0.2898 | −0.5615 |
| Machinery and Equipment Industry | −0.9632 | −1.8643 | −1.4225 | −2.7531 | −0.1144 | −0.2219 | −0.5711 | −1.1074 |
| Communications, Instruments Industry | −0.3051 | −0.5969 | −0.4506 | −0.8814 | −0.0864 | −0.1625 | −0.4310 | −0.8109 |
| Construction Industry | −2.0971 | −4.0370 | −3.0970 | −5.9618 | −0.0422 | −0.0818 | −0.2107 | −0.4082 |
| Transportation and Postal Industry | −2.4669 | −4.7364 | −3.6432 | −6.9948 | −0.0346 | −0.0670 | −0.1729 | −0.3342 |
| Service Industry | −1.3303 | −2.5677 | −1.9646 | −3.7920 | −0.0894 | −0.1733 | −0.4460 | −0.8649 |
| Coal Mining and Washing Industry | 0.2583 | 0.4987 | 0.3814 | 0.7365 | 0.0327 | 0.0633 | 0.1633 | 0.3160 |
| Coking Industry | −0.1987 | −0.3824 | −0.2934 | −0.5647 | −0.0783 | −0.1513 | −0.3906 | −0.7550 |
| Petroleum Extraction Industry | −0.9388 | −1.7885 | −1.3864 | −2.6413 | 0.0137 | 0.0254 | 0.0686 | 0.1266 |

**Table 5.** *Cont.*

| | Refined Oil Price Market Mechanism | | | | Gas Price Market Mechanism | | | |
|---|---|---|---|---|---|---|---|---|
| | Energy Consumption | | Carbon Emission | | Energy Consumption | | Carbon Emission | |
| | 5% | 10% | 5% | 10% | 5% | 10% | 5% | 10% |
| Petroleum Processing Industry | −0.8883 | −1.6861 | −1.3119 | −2.4902 | 0.0401 | 0.0778 | 0.2001 | 0.3881 |
| Natural Gas Extraction Industry | 0.8924 | 1.6990 | 1.3179 | 2.5091 | −0.1092 | −0.2108 | −0.5448 | −1.0519 |
| Gas Production Industry | 0.1713 | 0.3350 | 0.2530 | 0.4948 | −0.6962 | −1.3714 | −3.4742 | −6.8436 |
| Thermal Power | 0.1702 | 0.3296 | 0.2513 | 0.4866 | 0.0297 | 0.0577 | 0.1482 | 0.2877 |
| Clean Energy | 0.6005 | 1.1678 | | | 0.0488 | 0.0945 | | |

Table 5 clearly demonstrates that as China's economy expands, fueling a growing demand for petroleum, the increase in refined oil prices results in a significant reduction in carbon emissions across various sectors. Given that refined oil consumption substantially exceeds that of crude oil, the drop in carbon emissions triggered by rising refined oil prices is markedly more pronounced than that caused by crude oil price increases. Specifically, this reduction in carbon emissions is primarily observed in high-energy-consuming industries such as transportation, the chemical industry, and the metal smelting industry. A 5% hike in refined oil prices leads to carbon emission reductions of 3.64%, 1.91%, and 0.87% in these sectors, respectively. Other industrial sectors also see diverse levels of carbon emission decreases in response to rising refined oil prices, notably in agriculture, where the use of fertilizers, pesticides, and agricultural machinery—all reliant on refined oil—contributes to a significant reduction in carbon emissions. In the case of natural gas, which has a smaller share in the energy mix and lower carbon emissions, the decrease in carbon emissions resulting from price increases is considerably less pronounced compared to other energy sources. The impact of rising gas prices on carbon emissions is mainly concentrated in sectors like other mining industries, machinery and equipment manufacturing, and the service industry, where a 5% increase in gas prices leads to carbon emission reductions of 0.56%, 0.57%, and 0.45%, respectively.

4.4.3. Macroeconomic Variables and Household Welfare

Table 6 illustrates that due to the predominant demand for refined oil across various industrial sectors, increases in refined oil prices have a more substantial impact on the macroeconomy than hikes in crude oil prices. For instance, a 5% rise in refined oil prices results in income reductions of 0.014%, 0.076%, and 0.076% for households, enterprises, and the government, respectively, with the government and enterprises experiencing a notable drop in income. Key economic indicators such as investment, and imports/exports also see significant declines, with a 5% increase in refined oil prices causing decreases of 0.073%, 0.100%, and 0.092% in these areas, respectively, alongside a 0.31% fall in real GDP and a slight 0.01% dip in nominal GDP. The consumer price index climbs by 0.29%, markedly affecting household living standards, and household welfare drops by 392.98. Conversely, Table 6 shows that the repercussions of rising gas prices on the macroeconomy are considerably less severe than those of refined oil. A 5% uptick in gas prices results in minimal income decreases of 0.0013%, 0.0028%, and 0.0072% for households, enterprises, and the government, respectively. Other macroeconomic variables are marginally affected, with total investment shrinking by 0.0029%, total imports and exports contracting by 0.0069% and 0.0063%, respectively, and both real and nominal GDP decreasing by 0.0266% and 0.0004%, respectively. Exchange rates experience a minor increase of 0.02%, the overall consumer price index rises by 0.03%, and household welfare is reduced by 62.93.

**Table 6.** The influence of the oil and gas price fluctuations on macroeconomic quantity and the residents' welfare (%).

| | Refined Oil Price Market Mechanism | | Gas Price Market Mechanism | |
|---|---|---|---|---|
| | **5%** | **10%** | **5%** | **10%** |
| Residential Income | −0.0139 | −0.0278 | −0.0013 | −0.0024 |
| Residential Savings | −0.0139 | −0.0278 | −0.0013 | −0.0024 |
| Residential Taxes | −0.0139 | −0.0278 | −0.0013 | −0.0024 |
| Corporate Income | −0.0762 | −0.1474 | −0.0028 | −0.0056 |
| Corporate Savings | −0.0762 | −0.1474 | −0.0028 | −0.0056 |
| Corporate Taxes | −0.0762 | −0.1474 | −0.0028 | −0.0056 |
| Government Income | −0.0756 | −0.1499 | −0.0072 | −0.0138 |
| Total Investment | −0.0725 | −0.1405 | −0.0029 | −0.0058 |
| Total Savings | −0.0725 | −0.1405 | −0.0029 | −0.0058 |
| Total Imports | −0.1001 | −0.1808 | −0.0069 | −0.0134 |
| Total Exports | −0.0916 | −0.1660 | −0.0063 | −0.0121 |
| Real GDP | −0.3068 | −0.5971 | −0.0266 | −0.0512 |
| Nominal GDP | −0.0125 | −0.0235 | −0.0004 | −0.0008 |
| Interest Rate | −0.0808 | −0.1515 | 0.0000 | −0.0101 |
| Exchange Rate | 0.3737 | 0.7272 | 0.0202 | 0.0404 |
| Price Index | 0.2929 | 0.5757 | 0.0303 | 0.0505 |
| Residential Welfare | −392.9809 | −762.7217 | −62.9281 | −120.6647 |

*4.5. Impact of Electricity Price Fluctuations on the National Economy*

The core of electricity reform lies in the reform of electricity pricing. Despite the implementation of several electricity pricing policies in recent years to address energy conservation and environmental protection concerns, the current electricity sector reforms have been slow in achieving effective pricing mechanisms that can accurately reflect and regulate supply and demand dynamics. Therefore, expediting the marketization of electricity pricing and transitioning towards market-driven prices becomes crucial. Given the substantial differences in scale and policy between thermal power and clean energy power generation, this paper aims to simulate the effects of price increases in both types of electricity on the national economy.

4.5.1. Sectoral Price Output and Investment Consumption

Table 7 reveals that electricity, as a key driver of economic growth and the primary input factor for all industrial sectors, is deeply integrated into every facet of the economy. Therefore, any increase in electricity prices necessarily results in price hikes across all industrial sectors, rendering the impact of electricity price hikes "homogeneous" across different sectors and particularly pronounced compared to other energy sources. This is especially true for energy-intensive sectors. For example, in a scenario of a 5% thermal power price hike, prices in the mining, metal smelting, non-metallic mineral products, and chemical industries surge by 0.81%, 0.67%, 0.63%, and 0.55%, respectively, underscoring the critical significance of streamlining the electricity pricing system in reshaping China's industrial structure. Other industrial sectors also experience varying degrees of price increases in response to thermal power price hikes. As clean energy prices rise as a substitute, the increase in clean energy prices is also substantial, with a 1.68% spike resulting from thermal power price hikes.

Table 7 clearly illustrates that China's energy mix is heavily reliant on coal, with thermal power serving as the primary source of electricity. Consequently, the impact of thermal power price hikes on various industrial sectors is significantly greater than that of clean energy. The mechanism by which clean energy price increases drive up industrial sector prices mirrors that of thermal power, but the magnitude of price increases in each sector is lower than that of thermal power. For instance, a 5% increase in clean energy prices leads to price increases of 0.25%, 0.20%, 0.19%, and 0.17% in other mining industries,

metal smelting, non-metallic mineral products, and chemical industries, respectively, all of which are high-energy-consuming sectors. It is evident from Table 7 that the degree of output decline across various sectors resulting from clean energy price hikes is much smaller than that caused by thermal power. Conversely, when clean energy prices rise, it curbs the output of high-energy-consuming industries but boosts the output of thermal power due to energy substitution, with a 5% increase in clean energy prices triggering a 0.54% surge in thermal power output.

**Table 7.** The influence of thermal power and clean energy price fluctuations on the department's price and output (%).

| | Thermal Power Price Market Mechanism | | | | Clean Energy Market Mechanism | | | |
|---|---|---|---|---|---|---|---|---|
| | Sector Price | | Sector Output | | Sector Price | | Sector Output | |
| | 5% | 10% | 5% | 10% | 5% | 10% | 5% | 10% |
| Agriculture, Forestry, Animal Husbandry, and Fishery | 0.1515 | 0.2929 | −0.1059 | −0.2068 | 0.0505 | 0.0909 | −0.0295 | −0.0566 |
| Other Mining Industries | 0.8080 | 1.5857 | −0.9229 | −1.7976 | 0.2525 | 0.4747 | −0.2794 | −0.5370 |
| Food and Tobacco Industry | 0.2222 | 0.4242 | −0.1659 | −0.3238 | 0.0606 | 0.1212 | −0.0483 | −0.0926 |
| Textile Products Industry | 0.2727 | 0.5353 | −0.1153 | −0.2262 | 0.0808 | 0.1515 | −0.0428 | −0.0830 |
| Wood and Paper Industry | 0.4242 | 0.8181 | −0.3472 | −0.6779 | 0.1313 | 0.2424 | −0.0994 | −0.1908 |
| Chemical Industry | 0.5454 | 1.0706 | −0.4517 | −0.8790 | 0.1616 | 0.3232 | −0.1304 | −0.2503 |
| Non-Metal Mineral Industry | 0.6262 | 1.2221 | −0.5127 | −0.9991 | 0.1919 | 0.3636 | −0.1585 | −0.3047 |
| Metal Smelting Industry | 0.6666 | 1.3029 | −0.6076 | −1.1830 | 0.2020 | 0.3939 | −0.1781 | −0.3420 |
| Machinery and Equipment Industry | 0.4545 | 0.8888 | −0.5651 | −1.1013 | 0.1414 | 0.2626 | −0.1695 | −0.3253 |
| Communications, Instruments Industry | 0.3333 | 0.6464 | −0.3040 | −0.6230 | 0.1010 | 0.1919 | −0.0793 | −0.1600 |
| Construction Industry | 0.4040 | 0.7878 | −0.4808 | −0.9378 | 0.1212 | 0.2323 | −0.1555 | −0.2997 |
| Transportation and Postal Industry | 0.2424 | 0.4646 | −0.2843 | −0.5538 | 0.0707 | 0.1313 | −0.0744 | −0.1423 |
| Service Industry | 0.1616 | 0.3232 | −0.1879 | −0.3655 | 0.0505 | 0.0909 | −0.0518 | −0.0990 |
| Coal Mining and Washing Industry | 0.3636 | 0.7070 | −0.9982 | −1.9312 | 0.1111 | 0.2121 | 0.6768 | 1.3071 |
| Coking Industry | 0.4747 | 0.9393 | −0.4334 | −0.8454 | 0.1414 | 0.2828 | −0.1539 | −0.2972 |
| Petroleum Extraction Industry | 0.4848 | 0.9494 | 0.2763 | 0.5335 | 0.1414 | 0.2828 | 0.1543 | 0.2982 |
| Petroleum Processing Industry | 0.4444 | 0.8686 | 0.3093 | 0.6007 | 0.1313 | 0.2525 | 0.1564 | 0.3024 |
| Natural Gas Extraction Industry | 0.4848 | 0.9494 | 0.2752 | 0.5314 | 0.1414 | 0.2828 | 0.1540 | 0.2975 |
| Gas Production Industry | 0.3939 | 0.7777 | 0.2167 | 0.4233 | 0.1212 | 0.2323 | 0.0985 | 0.1905 |
| Thermal Power | 5.0500 | 10.1000 | −4.6934 | −8.9944 | 0.5252 | 1.0201 | 0.5382 | 1.0326 |
| Clean Energy | 1.6766 | 3.3027 | 1.6680 | 3.2406 | 5.0500 | 10.1000 | −7.8354 | −14.7925 |

### 4.5.2. Sectoral Energy Consumption and Carbon Emissions

Table 8 provides clear evidence that thermal power is the primary energy source for various industrial sectors. The increase in thermal power prices has a significant and wide-ranging impact on these sectors. In the case of a 5% price hike, the effect on energy consumption exceeds 1% in most sectors, except for those that undergo energy substitution. Heavy industries are particularly affected. For instance, other mining industries, metal smelting, and machinery equipment manufacturing witness a reduction of 2.16%, 1.45%, and 2.06% in energy consumption, respectively. The impact on energy consumption is also substantial in other sectors, even in lower energy-consuming areas such as agriculture and services, which experience decreases of 1.76% and 1.61% in energy demand, respectively. As a result of the substitution effect, the increase in electricity prices leads to a higher

demand for alternative energy sources. For example, a 5% increase in thermal power prices corresponds to a 0.27% increase in energy consumption in petroleum processing and a 0.11% increase in gas production industries. Similarly, rising clean energy prices also lead to reduced energy consumption across different industrial sectors, affecting heavy industries more significantly than light industries, agriculture, and services. For instance, a 5% increase in clean energy prices results in a 0.67% decrease in energy consumption in other mining industries, 0.56% in agriculture, and 0.50% in services.

**Table 8.** The influence of the thermal power and clean energy price fluctuations on the department's energy consumption and energy intensity (%).

| | Thermal Power Price Market Mechanism | | | | Clean Energy Market Mechanism | | | |
| --- | --- | --- | --- | --- | --- | --- | --- | --- |
| | Energy Consumption | | Carbon Emission | | Energy Consumption | | Carbon Emission | |
| | 5% | 10% | 5% | 10% | 5% | 10% | 5% | 10% |
| Agriculture, Forestry, Animal Husbandry, and Fishery | −1.7662 | −3.4146 | −0.8639 | −1.6881 | −0.5558 | −1.0669 | 0.2833 | 0.5466 |
| Other Mining Industries | −2.1627 | −4.1760 | −1.4847 | −2.7129 | −0.6779 | −1.3007 | 0.6234 | 1.1729 |
| Food and Tobacco Industry | −1.7490 | −3.3810 | −1.1293 | −2.2903 | −0.5539 | −1.0631 | 0.3084 | 0.6693 |
| Textile Products Industry | −1.4827 | −2.8680 | −0.9199 | −1.8338 | −0.4656 | −0.8939 | 0.3823 | 0.6654 |
| Wood and Paper Industry | −1.8671 | −3.6080 | −1.1771 | −2.3057 | −0.5855 | −1.1235 | 0.3846 | 0.7426 |
| Chemical Industry | −2.0905 | −4.0132 | −2.2210 | −3.9183 | −0.3313 | −0.6360 | 0.7300 | 1.3206 |
| Non-Metal Mineral Industry | −2.1639 | −4.1559 | −2.2384 | −4.3349 | −0.3636 | −0.6986 | 1.0274 | 2.0282 |
| Metal Smelting Industry | −2.0526 | −3.9109 | −2.2096 | −4.3261 | −0.4468 | −0.8575 | 1.1465 | 1.9834 |
| Machinery and Equipment Industry | −2.0586 | −3.9754 | −1.5498 | −2.8681 | −0.6479 | −1.2429 | 0.7851 | 1.2398 |
| Communications, Instruments Industry | −1.6972 | −3.2843 | −1.5072 | −2.5928 | −0.5247 | −1.0070 | 0.4724 | 0.7780 |
| Construction Industry | −1.4260 | −2.7620 | −1.5144 | −2.8430 | −0.4560 | −0.8761 | 0.4402 | 0.7852 |
| Transportation and Postal Industry | −0.5682 | −1.1049 | −1.6848 | −3.1948 | −0.1644 | −0.3153 | 0.6695 | 1.2411 |
| Service Industry | −1.6060 | −3.1058 | −0.8593 | −1.7011 | −0.5043 | −0.9677 | 0.2672 | 0.5357 |
| Coal Mining and Washing Industry | −0.5973 | −1.1453 | −0.6293 | −1.1902 | 0.5502 | 1.0621 | 0.3468 | 0.6462 |
| Coking Industry | −0.2874 | −0.5602 | −1.7181 | −3.4109 | −0.1072 | −0.2072 | 0.4376 | 0.7647 |
| Petroleum Extraction Industry | −0.9464 | −1.8400 | −1.5612 | −3.0593 | −0.2363 | −0.4534 | 0.5600 | 1.0819 |
| Petroleum Processing Industry | 0.2702 | 0.5242 | 1.3569 | 2.6940 | 0.1440 | 0.2785 | 0.4714 | 0.8304 |
| Natural Gas Extraction Industry | −0.9474 | −1.8420 | −1.2573 | −2.0474 | −0.2366 | −0.4541 | 0.5598 | 1.0814 |
| Gas Production Industry | 0.1123 | 0.2193 | 0.4970 | 0.8444 | 0.0657 | 0.1273 | 0.2491 | 0.4437 |
| Thermal Power | −1.4536 | −2.8647 | −2.1994 | −4.1119 | 0.3796 | 0.7262 | 1.1999 | 2.0944 |
| Clean Energy | 0.0887 | 0.1431 | | | −3.3294 | −6.4567 | | |

According to Table 8, the increase in thermal power prices results in reduced energy consumption across various sectors. However, the decline in carbon emissions is not as significant as that caused by coal due to energy substitution. Nevertheless, in high-energy-consuming sectors with substantial electricity demand, the decrease in carbon emissions due to higher thermal power prices is more pronounced. For instance, in other mining industries, the chemical industry, non-metallic mineral processing industry, and metal smelting industry, a 5% increase in thermal power prices leads to carbon emission reductions of 1.48%, 2.21%, 2.24%, and 2.20%, respectively. Conversely, in sectors like agriculture and services, the decline in carbon emissions resulting from increased thermal power prices is much smaller compared to the high-energy-consuming industries. On the other hand, with the rise in clean energy prices, there is an upsurge in the utilization of thermal

power and fossil fuels across sectors, which are the primary sources of carbon emissions. Consequently, the increase in clean energy prices leads to higher carbon emissions across sectors. This effect is particularly noticeable in high-energy-consuming industries due to their significant energy demands. For instance, a 5% increase in clean energy prices leads to carbon emission increases and higher carbon intensity in the non-metallic mineral processing industry, metal smelting industry, and thermal power sector by 1.03%, 1.15%, 1.20%, and 1.36%, 1.26%, 1.72%, respectively.

### 4.5.3. Macroeconomic Variables and Household Welfare

According to Table 9, the increase in thermal power prices has a significant impact on business production and household living. Compared to other energy sources, it has the largest economic consequences. With a 5% price increase, residents, businesses, and the government experience a decrease in income by 0.019%, 0.097%, and 0.114%, respectively. Total investment and savings decline by 0.086%, while trade is significantly affected, with a decrease of 0.208% and 0.185% in total imports and exports. The overall economy suffers as real GDP and nominal GDP decline by 0.316% and 0.011%, respectively. Furthermore, capital returns decrease, interest rates fall by 0.10%, exchange rates rise by 0.39%, the consumer price index increases by 0.30%, and household welfare decreases by 403.01. In contrast, the impact of rising clean energy prices on the macroeconomy is smaller due to its smaller proportion in China's power structure. A 5% increase in clean energy prices results in a 0.004%, 0.044%, and 0.02% reduction in income for residents, businesses, and the government, respectively. Overall output is minimally affected, with a 0.11% decrease in real GDP and a 0.01% decrease in nominal GDP. Interest rates fall by 0.044%, exchange rates rise by 0.12%, the consumer price index increases by 0.091%, and household welfare decreases by 122.48, according to Table 9.

**Table 9.** The influence of the thermal power and clean energy price fluctuations on macroeconomic quantity and the residents' welfare (%).

| | Thermal Power Price Market Mechanism | | Clean Energy Market Mechanism | |
|---|---|---|---|---|
| | **5%** | **10%** | **5%** | **10%** |
| Residential Income | −0.0186 | −0.0364 | −0.0044 | −0.0086 |
| Residential Savings | −0.0186 | −0.0364 | −0.0044 | −0.0086 |
| Residential Taxes | −0.0186 | −0.0364 | −0.0044 | −0.0086 |
| Corporate Income | −0.0966 | −0.1896 | −0.0440 | −0.0854 |
| Corporate Savings | −0.0966 | −0.1896 | −0.0440 | −0.0854 |
| Corporate Taxes | −0.0966 | −0.1896 | −0.0440 | −0.0854 |
| Government Income | −0.1138 | −0.2224 | −0.0278 | −0.0535 |
| Total Investment | −0.0861 | −0.1689 | −0.0367 | −0.0711 |
| Total Savings | −0.0861 | −0.1689 | −0.0367 | −0.0711 |
| Total Imports | −0.2084 | −0.4048 | −0.0482 | −0.0917 |
| Total Exports | −0.1853 | −0.3602 | −0.0434 | −0.0827 |
| Real GDP | −0.3163 | −0.6169 | −0.1009 | −0.1942 |
| Nominal GDP | −0.0114 | −0.0229 | −0.0100 | −0.0194 |
| Interest Rate | −0.1010 | −0.1919 | −0.0404 | −0.0808 |
| Exchange Rate | 0.3939 | 0.7777 | 0.1212 | 0.2323 |
| Price Index | 0.3030 | 0.5959 | 0.0909 | 0.1717 |
| Residential Welfare | −403.0082 | −782.0834 | −122.4827 | −235.1280 |

## 5. Discussion

By simulating exogenous increases of 5% and 10% in different energy prices, it becomes evident that while energy price hikes lead to a reduction in total GDP and labor remuneration, they also drive adjustments in industrial structure and spur technological innovation. For instance, a 5% and 10% increase in coal prices results in a GDP decrease of 0.23% and 0.45%, respectively, accompanied by an energy demand reduction of 0.80%

and 1.54%. Rising energy prices accelerate the transition towards greener technologies and facilitate economic growth recovery. Additionally, they contribute to environmental pollution reduction and lower carbon emissions. Considering the transmission mechanism within the industrial chain, the rise in energy prices does incentivize energy conservation and emission reduction, albeit different from a direct decline in carbon emissions. Although various industries, particularly heavily polluting sectors, experience a negative impact on output due to higher energy prices, this mechanism ensures that enterprises are motivated to innovate in green technology, boost investments in environmentally friendly industries, and foster high-quality economic development. Moreover, the market-oriented approach to energy prices has led to increased output of clean power, enhancing its competitive advantage relative to the decline in thermal power production. Based on these findings, the following conclusions and policy suggestions can be drawn.

1. Increasing fossil energy prices leads to decreased demand for fossil fuels and increased demand for clean electricity in China, which is heavily reliant on coal and oil. This results in reduced overall energy consumption and improved energy efficiency.
2. Higher fossil energy prices contribute to a decrease in China's total carbon emissions. Market-based electricity pricing mechanisms are more effective than government regulation in reducing carbon emissions. Reforming the electricity market helps transmit price signals within the economy, leading to emissions reduction.
3. Competitive pricing of clean electricity compared to traditional thermal power stimulates demand for clean energy. Market-driven electricity pricing mechanisms play a significant role in promoting clean electricity consumption. Government regulation of electricity prices also supports the development of clean energy. Therefore, reducing the cost of clean electricity to ensure price competitiveness with thermal power is crucial for advancing clean and renewable energy sources.

The primary limitation of this study is the challenge in accurately quantifying the administrative costs associated with government intervention in the energy market, an aspect not accounted for in our analysis. This gap presents an opportunity for future research to explore. Moreover, given that the CGE model largely depends on theoretical frameworks and lacks the capacity to delineate the effects of energy price fluctuations on economic growth and the energy environment using historical data, our next steps will involve leveraging econometric models based on historical data to investigate the impacts of changes in energy prices on economic development and the energy landscape.

**Author Contributions:** Methodology, Y.G.; Writing—original draft preparation, L.Y. All authors have read and agreed to the published version of the manuscript.

**Funding:** This research received no external funding.

**Institutional Review Board Statement:** Not applicable.

**Informed Consent Statement:** Not applicable.

**Data Availability Statement:** The data presented in this study are available on request from the corresponding author.

**Conflicts of Interest:** The authors declare no conflicts of interest.

## Appendix A

*Appendix A.1 CGE Modeling*

Appendix A.1.1 Production Module

The production module of this study adopts a nested structure comprising seven levels. At the first level, total output is derived from a combination of the capital-energy-labor synthetic bundle and intermediate inputs. The CES production function is employed, with total intermediate inputs represented as a Leontief synthesis across sectors, indicating proportional allocation of each sector's intermediate inputs within the overall framework.

Moving to the second level, the capital-energy-labor synthetic bundle is further decomposed into labor and capital-energy synthetic bundles, utilizing the CES production function. The third level of nesting then breaks down the capital-energy synthetic bundle into its constituent elements: capital and energy, once again employing the CES production function. The energy component is subject to a fourth level of nesting, encompassing a CES synthesis of electricity energy and fossil energy. Within the fifth level of nesting, electricity is subdivided into thermal and other electricity, while fossil energy is divided into coal and oil and gas. Hydrocarbons, existing at the sixth layer of the production structure, represent the synthetic combination of oil and natural gas. Further dissection reveals coal as raw coal and coke, oil as oil extraction and oil processing, and natural gas as natural gas extraction and natural gas processing.

Appendix A.1.2 Price Module

The structure of the price module mirrors that of the production module, with a layered synthesis that establishes interdependencies among sectors. This ensures that fluctuations in energy prices, as a crucial upstream factor in the industrial chain, impact price variations and outputs across sectors. Within the price module, this study assumes that the country under examination constitutes only a small portion of the global economy, with domestic commodity prices having no influence on international market prices. Prices within the module are relative prices, and a CES function is employed to link domestic and foreign products, incorporating factors such as taxes, exchange rates, and other interconnections.

Appendix A.1.3 Income Expenditure Module

The income-expenditure module outlines the earnings and spending patterns of individuals, businesses, and the government. Residents' income comprises compensation received from firms in exchange for their labor, along with transfers from both firms and the government. Meanwhile, residents' expenditures encompass the savings portion as well as their consumption of goods. Government revenue consists of all taxes collected by the government sector, including both direct and indirect taxes. Enterprises derive their income from capital gains.

Appendix A.1.4 The Trade Module

Within the trade module, goods are categorized into three groups: domestically produced goods for export, domestically produced goods for domestic consumption, and imported goods. There exist both similarities and differences between domestically produced goods and imported/exported goods, indicating imperfect substitutability. Moreover, there is also imperfect substitution between goods destined for export and those intended for domestic sale. The module assumes that producers must determine the proportion of goods to supply in the domestic and international markets, tailor the goods to suit the characteristics of each target market, and ultimately transport the goods to both markets.

Appendix A.1.5 Carbon Emissions Module

Energy consumption is a value-based metric, meaning that carbon emissions cannot be calculated based on physical quantities alone. Instead, corresponding carbon emission coefficients for the value-based measure are utilized for calculation purposes. Firstly, carbon emissions must be computed based on the physical quantity aligned with the social accounting matrix. Notably, since refined oil products such as fuel oil, gasoline, kerosene, and diesel have comparable carbon emission coefficients, the mean value of the four is used to calculate emissions. Once the carbon emission coefficients for each energy type are obtained, the value-based carbon emission coefficient is established by comparing the final demand associated with the social accounting matrix.

Appendix A.1.6 Equalization Module

The equilibrium module is essential to ensure the solvability of the CGE model and includes three market clearances and three macro-equilibria. The three market clearances pertain to commodity, labor, and capital markets, respectively. The three macro-equilibria refer to the balance of payments, government budget, and investment and savings equilibrium.

*Appendix A.2 SAM Table Construction*

The primary data source for compiling the SAM table is the China Input-Output Table (2020). Additional data from statistical yearbooks, financial yearbooks, and energy statistics yearbooks are integrated to supplement the available information. In cases where exact sources cannot be found, residual items or proportional distribution methods are utilized. Specifically, data on intermediate inputs, labor compensation, return on capital, residential consumption, government consumption, investment, and imports are obtained from the 2020 China Input-Output Table. Residential savings, corporate savings, and government savings are derived from the China Statistical Yearbook (2021), while personal income tax data is sourced from the China Economic Yearbook (2021).

The macro SAM table lacks industry-specific divisions to analyze the impact of economic shocks on each sector. To address this, we developed a more detailed micro SAM table by dividing the input-output table of 139 sectors into 21 sectors through merging and decomposition. In addition to sector merging, we also subdivided individual sectors for this study. For example, we divided the "oil and gas extraction products" sector into separate "oil" and "gas" sectors based on the energy consumption proportions derived from the input-output table. Similarly, the "Electricity and heat production and supply" sector was divided into "Thermal power" and "Other power" sectors using data from the China Energy Statistics Yearbook (2021) and the 2012 Power Balance Sheet within the same yearbook. The industrial sectors included in the micro SAM table are: agriculture (01), other mining (02), food (03), textile (04), wood and paper (05), chemical industry (06), non-metallic mining (07), metal smelting (08), machinery and equipment (09), communications and instrumentation (10), construction (11), transportation and postal services (12), services (13), coal and washing (14), coal and oil (15), coking (16), petroleum extraction (17), petroleum processing (18), natural gas extraction (19), gas production (20), thermal power (21), and clean energy (21).

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
