# Peer review of "Marketization of Energy Resources in China: An Environmental CGE Analysis"

_sustainability, doi:10.3390/su16041463_

Round 1
Reviewer 1 Report
Comments and Suggestions for Authors
See the attached report. The authors should give more credit to the original developers of the ideas and techniques that they cite as opposed to citing only recent works by Chinese authors.

The English is fine, but the prose itself a bit wordy and the sentences are typically over-long.
Author Response
Thank you very much for carefully reading the manuscript. We are very grateful for your helpful and valuable recommendations for further improvements to the paper. We have revised the manuscript based on the reviewers’ comments and suggestions, which is shown in the Word file.

Reviewer 2 Report
Comments and Suggestions for Authors
Author Response

(The authors gave the same response as above.)

Round 2
Reviewer 1 Report
Comments and Suggestions for Authors
Manuscript Review of Sustainability-2809258-v2
Research on the Impact of Energy Price Fluctuations on the Energy-Environment-Economic System: An Analysis Based on the CGE Model
Li Yang, Ya Gao
Overview. This is my second review of this manuscript. The modeling system is untouched but is described more thoroughly, including the model’s sensitivity to key elasticities. The manuscript’s purpose is immediately clearer. Its conclusions now appear to be more modest, but not very interesting:
1) Energy price rises cause systemwide domestic price rises within China, which dampen demand for Chinese products and services but also reduce CO2 emissions;
2) Chinese government regulation of electricity prices mitigates some of the aforementioned deleterious effects of energy price rises but thereby also enables CO2 emissions;
3) Marketization of electricity (and its use as opposed to, say, thermal heating) promotes cleaner energy policy.
Review. The bottom line is that the conclusions remain quite boring. Such outcomes naturally emerge from a (more) market-oriented economy. It is not clear why the complex modeling system developed herein might be required to attain them. A far simpler DSGE model (e.g., Bondzie, Di Bartolomeo, & Fosu, 2014; Xiao, Fan, & Guo, 2018; Punzi, 2019) would undoubtedly lead to similar findings. The authors’ use of a 21-sector CGE model suggests the resulting research will feature a display and discuss the rich set of intersectoral relationships availed by such a model, something not evident in the present version of this manuscript.
The start of the manuscript remains tedious. In part, this is because Introduction’s sentences tend to be overly long—the first few sentences, in particular. Worse, price fluctuations in China and the causes of them are not mentioned until the middle of the first paragraph (a four-line sentence that starts on line 34); although at least they are now mentioned, unlike in the prior rendition of the manuscript. The reader needs to know what policy problem the authors’ CGE model will address and why. This is finally suggested in the sentence starting on line 38. “[E]nergy prices are still largely under government control [in China] and[, therefore] cannot fully reflect the scarcity of energy resources …[3].” This should be the paper’s first sentence. The material before this sentence is undoubtedly true, but it isn’t core to the manuscript’s primary premise. It’s not like the manuscript investigates economic distortions beyond energy prices like “market segmentation, government regulation, monopolistic forces, and other factors, the prices of production factors like capital, labor, … in China…”. Why not?
The authors report that literature “on the of energy price distortions…confirms that energy prices affect environmental pollution and economic growth[5,6].” But this is the limit of what they conclude in this manuscript as well. This similarity in findings suggests that they must be using a better approach (that yields deeper and more conclusive insight) than any previously used in some way. They tell the reader that novelties of the research in the present manuscript are that they separate energy resources production and use from electricity production and use. But such detail has been modeled by input-output and CGE models in China previously, albeit perhaps with as much detail on energy resources. The literature review clearly needs to be contained within the Introduction; otherwise, readers are left waiting to find out why they should bother reading the remainder of the manuscript.
We, Sustainability’s international readership, still don’t know why we should care about research on such matter in China. This needs to be established somewhere in the Introduction.
Readers still can’t know what scientific test is undertaken in the research reported in the manuscript. Theory alone can tell us how an economy that is more market-oriented is more sensitive to prices, and that prices regulate use (demand) and production levels. So, what sort of special insight is the authors’ CGE expected beyond that a relaxation of energy product prices will enable a regulation their use via the market’s “invisible hand”.
Do the authors remove the administrative costs (to industry and government) of government’s interference in energy markets? If they do, they do not identify those costs or the mechanism by which they reduce them.
It is now clear that the authors assume neoclassical model closure. Is this reasonable for electric power production? Even with reform won’t they at least remain spatial monopolies and not exist in a perfectly competitive market required by neoclassical closure? I really doubt that any aspect of the energy industry is completely competitive “between the labor market and the capital market, with no unemployment and no surplus capital” (line 185-187). That is, the authors merely assert that (line 182-184) “this paper considers the use of the neoclassical macroeconomic closure to be appropriate.” They need to give a better rationale for this since power production, at least, remains heavily regulated in the US and the EU, for example.
The reader finally learns at line 205 that the authors will employ different policy scenarios built on different fossil energy-price ranges and on different reforms for the electricity marketization. The present manuscript is not only unclear about what they are, but totally fails to inform the reader about why they might be interesting or worthy of investigation. Such rationales should be established via material discussed in the Introduction. And the scenarios should be introduced, outlined, and rationalized at the start of what is now Section 4 Policy Simulation.
Unfortunately, the Introduction does not presently discuss China’s relatively heavy (compared to other nations) reliance on coal—especially for electric power production—or coal’s tendency to generate more pollutants compared to other fossil fuels. Moreover, it seems the costs of transforming toward a more fossil-fuel free economy is costless within the modeling system. Certainly, this is not the case. Nothing is cost free, and even solar and wind power have substantial environmental costs associated with them. There is no discussion about or reference to such matter.
Who will pay for the technological transformation? How much will it cost? What is the alternative resource? If it is a fossil fuel, where will it come from (crude oil is not abundant in China)? What is the CO2 difference when oil or natural gas is burned instead of coal? If the alternative resources are non-hydroelectric and yet renewable, how will storage of electricity (when the wind and/or the solar power generation are unavailable) be facilitated? What are the costs of such storage? What are the modeling mechanisms that enable endogenize the costs of the alternatives? None of this is clear at present.
Won’t different regional outlooks for future electricity demand within China matter? Won’t residential demand for temperature comfort in China’s southeast put more pressure on the China’s electric power grid than the same sort of residential demand in the nation’s northeast? What are each scenario’s underlying assumptions for regional population and industry growth in China?
While a discussion of results rarely makes for a scintillating read, the current version is overly detailed. This makes it exceptionally boring and tedious. The authors should reduce the discussion to those items that are most novel and important, in that order of importance. Reducing the discussion or results will make room for them to elaborate elsewhere on matter of more interest and critical importance to the reader.
The conclusions need to be something other than what the authors focus upon now. By how much do the authors let exogenous changes in the energy resources prices vary? How much would this reduce aggregate annual administrative costs that are presently associated with current levels of regulation? By how much would such price and administrative changes likely alter aggregate GDP, labor compensation, and carbon emissions nationwide? What are the investment costs of such an energy market transition? What incentives would be engaged to assure such technological transition occurs? Outside of the energy industry, what industries will win and lose most (this is the true benefit of a CGE model)?
REFERENCES
Bondzie, Eric, Giovanni Di Bartolomeo, and Gabriel Obed Fosu. (2014). “Oil Price Fluctuations and it Impact on Economic Growth: A DSGE Approach,” Available at SSRN 2729833 http://dx.doi.org/10.2139/ssrn.2729833.
Punzi, Maria Teresa. (2019). “The impact of energy price uncertainty on macroeconomic variables,” Energy Policy, 129, 1306–1319.
Xiao, Bowen, Ying Fan, and Xiaodan Guo. (2018). “Exploring the macroeconomic fluctuations under different environmental policies in China: A DSGE approach,” Energy Economics, 76, 439–456.
Comments on the Quality of English LanguageThe prose is readable, yet tedious to read. Break up the sentences.
Author Response
Thank you very much for carefully reading the manuscript. We are very grateful for your helpful and valuable recommendations for further improvements to the paper. We have revised the manuscript based on the reviewers’ comments and suggestions, which are shown in the WORD.

Reviewer 2 Report
Comments and Suggestions for Authors
After the corrections you have made the manuscript is eligibl;a for publication.
Author Response
Thank you.
Round 3
Reviewer 1 Report
Comments and Suggestions for Authors
Manuscript Review of Sustainability-2809258-v3
Research on the Impact of Energy Price Fluctuations on the Energy-Environment-Economic System: An Analysis Based on the CGE Model
Li Yang, Ya Gao
Overview. This is my third review of this manuscript. The purpose and mode of analysis is now clear. Its conclusions remain general and quite uninteresting—essentially only reporting findings that would arise from a theoretical model, not an empirical model with industry detail.
Review. As I reported before, the conclusions are boring. Thus, the paper inadvertently suggests that no one should bother to take the effort to build and use a 21-sector CGE model of China. This probably is not the message that the authors want to convey. So, the menuscript still needs some improvements; let’s hope it is one last round.
As it seems the manuscript is converging toward publication quality, it is time to focus on creating a crisper, catchier title. It is now overly long. Let me suggest “Marketization of Energy Resources in China: An Environmental CGE Analysis.”
The manuscript’s first two sentences mention an “energy transition,” but presently no discussion exists of why an energy transition might transpire in China. As a result, I suggest the first sentence be altered to be something more like (borrowing from the start of Section 4):
Energy resource markets in China have been liberalizing. Its secondary energy markets, however, largely remain under government control. This transitional experience can yield valuable insight, not only on-going change in China but also for other industrializing nations that are not yet fully market oriented.
But even this is insufficient since most readers might not know what “secondary energy markets” might be. With this starting point, the rest of the Introduction’s first paragraph can be rewritten more efficiently and effectively to home in upon the paper’s main thesis. The Introduction and Literature Review continue to cover a broad swath of noncritical material that neither motivates nor informs the performed research well. I continue to suggest (strongly) that these first two sections be rewritten as one, with an eye toward reducing irrelevant side stories (mentioning VAR, SVAR, and ECMs, for example).
On page 3, the authors write:
…econometric models have high requirements for historical data, necessitating the use of stable past data. Therefore, in the data processing process, many important pieces of information are lost. Empirical test results are quite sensitive to the selection of variables and the subjective setting of models. Moreover, econometric models only provide a trend-based description when analyzing the impact of energy price fluctuations on the economy. Therefore, using CGE models for research is more suitable for modeling the impact of energy prices[15,16].
The second and third sentence of the above are strong assertions and, from my perspective, untrue, at least for econometric/input-output models like those used by Kratena and colleagues. This is because econometric/input-output models often articulate the economy in more industry detail than does the manuscript I am reviewing. Moreover, the review of research immediately preceding the above lines is strictly a list. It should, instead, providing the reader with details pertaining to the present manuscript’s thesis—the mechanism(s) used to enable the analysis of price changes on the environment and economy and the CGE model’s sectoral detail (as opposed to the lack thereof in DSGEs). Ultimately the key to the authors’ argument militating for the use of CGE models is that they enable economic change that is not described via historical data by relying, instead, on changes guided by theory. (They’ll need a reference for this.) The application of theory rather than empirical history is important for analyzing economic disturbances previously not experienced by an economy and for examining changes in economic regime, like moving toward a market economy from one dominated by government ownership/management.
I remain unconvinced by the authors’ argument for neoclassical model closure. But I relent on this matter, provided they appropriately and thoroughly address all other issues that I’ve raised.
The authors suggest that the market will assure the requisite technological transformation. Both Europe and the US are subsidizing theirs to quicken the pace of transition. How long will China’s transformation to fully marketized energy take without national government subsidies (intervention)? Time matters!
The conclusions need to become more interesting. By how much does the economywide demand for fossil fuels change in the long run with energy price changes of 5 and 10%? Economywide GDP? Do emissions fall by as much as energy prices rise? Why or why not? How can the government reduce the cost of clean electricity to make it price-competitive with thermal power without “intervening” in energy markets?
Comments on the Quality of English LanguageSentences still tend to be too long. Many long sentences, end to end, make the reading rather tedious.
Author Response

(The authors gave the same response as above.)
